

# Remapping precipitation in mountainous area based on vegetation pattern

Xing Zhou[1], Guang-Heng Ni[1], Chen Shen[1], Ting Sun[1]

1) State Key Laboratory of Hydro-Science and Engineering, Department of Hydraulic Engineering, Tsinghua University, Beijing 100084, China

*Corresponding to:* Ting Sun( sunting@tsinghua.edu.cn )

**Abstract.** Accurate high-resolution estimates of precipitation are vital to improve the understanding on basin-scale hydrology in mountainous areas. The traditional interpolation methods or satellite-based remote sensing products are known to have limitations in capturing spatial variability of precipitation in mountainous areas. In this study, we develop a fusion framework to improve the precipitation estimation in mountainous areas by jointly utilizing the satellite-based precipitation, gauge measured precipitation and vegetation index. The development consists of vegetation data merging, vegetation response establishment, and precipitation remapping. The framework is then applied to the mountainous area of Nu River basin for precipitation estimation. The results demonstrate the reliability of the framework in reproducing the high-resolution precipitation regime and capturing its high spatial variability in Nu River basin. In addition, the framework can significantly reduce the errors in precipitation estimates as compared with the inverse distance weighted (IDW) method and TRMM (Tropical Rainfall Measuring Mission) precipitation product.

## 1 Introduction

Precipitation plays an important role in hydrological process, land-atmospheric processes, and ecological dynamics. Accurate high-resolution precipitation is crucial for streamflow prediction, flood control, and water resources management in data-sparse regions such as mountainous areas (Song et al., 2015). However, it is of great challenge to obtain accurate precipitation in mountainous areas due to the sparse gauge network and the remarkable spatiotemporal variability of precipitation. Conventional gauge networks can provide accurate rainfall measurements at point scales, which can be interpolated within the region of interest to give estimates of





precipitation in ungauged areas. However, such interpolated estimates might not be reliable in mountainous areas
considering the very limited gauges there (Phillips et al., 1992; Mair and Fares, 2011; Jacquin and Soto-Sandoval,
2013; Wang et al., 2014; Borges et al., 2016).

Recently, remote-sensing-based precipitation (RSBP) products, such as the Global Precipitation Climatology
Project (GPCP) (Schamm et al., 2014), the Tropical Rainfall Measuring Mission (TRMM) (Council, 2005), and
the Climate Prediction Center Morphing Method (CMORPH) (Joyce et al., 2004) so on, have been extensively
used in ungauged or sparsely-gauged areas to bridge the gap between the need for precipitation estimate and the
scarcity in gauge observations (Akbari et al., 2012; Kneis et al., 2014;  Li et al., 2015; Worqlul et al., 2015;
Mourre et al., 2016; Wong et al., 2016). Also, data fusion across satellite and gauge observations is being
conducted to further the application of RSBPs ( Rozante et al., 2010; Woldemeskel et al., 2013; Arias-Hidalgo et
al., 2013; Chen et al., 2016; Zhou et al., 2016). However, due to the relatively coarse spatial resolution (e.g.,
0.25°–5°) and uncertainties of RSBPs, their applications in mountainous basins, where the precipitation shows
large spatial variability, are still very limited ( Krakauer et al., 2013; Chen and Li, 2016).

Precipitation estimates can be influenced by a variety of ambient factors (e.g., topography, vegetation, etc.). In
order to correct effects of topography in precipitation estimate, Digital Elevation Model (DEM) has been widely
used in spatial interpolation of precipitation over mountainous areas (Marquínez et al., 2003; Lloyd, 2005).
However, the relationship between elevation and precipitation is not clear. Meanwhile, strong correlations
between NDVI and precipitation are found by several studies (Li et al., 2002; Kariyeva and Van Leeuwen, 2011;
Li and Guo, 2012; Sun et al., 2013; Campo-Bescós et al., 2013). As such, establishing statistical models between
normalized difference vegetation index (NDVI) and precipitation so as to improve the spatial resolution of
TRMM products in mountainous areas is becoming popular (Immerzeel et al., 2009; Jia et al., 2011; Duan and
Bastiaanssen, 2013; Chen et al., 2014; Xu et al., 2015; Mahmud et al., 2015;  Jing et al., 2016). For instance,
Immerzeel et al. (Immerzeel et al., 2009) downscaled TRMM-3B43 to 1 km based on an exponential relationship
between NDVI and TRMM precipitation in Iberian Peninsula of Europe. Jia et al. (Jia et al., 2011) established four
multivariable linear regression models between TRMM-3B43 precipitation and two other factors (i.e., DEM and
NDVI) of different resolutions (0.25°, 0.5°, 0.75°, 0.1°) to get 1 km estimates of precipitation in the Qaidam Basin
of China. Duan and Bastiaanssen (Duan and Bastiaanssen, 2013) used nonlinear relationship between



TRMM-3B43 and NDVI to downscale precipitation to 1 km in a humid area and a semi-arid area. Chen et al.
(Chen et al., 2014) established spatially varying relationship among TRMM, NDVI, and DEM by using a local
regression analysis approach known as geographically weighted regression (GWR) in South Korea. Xu et al. (Xu
et al., 2015) also used the GWR method to explore the spatial heterogeneity of the RSBP-NDVI and RSBP-DEM
relationships over two mountainous area in western China.

However, the present RSBP-NDVI-based schemes have several limitations: 1) significant errors can be
introduced during the downscaling given the nonlinear relationship between RSBP and NDVI; 2) large
uncertainties exist in the RSBP for mountainous areas, and 3) inter-comparison of existing NDVI datasets are
missing in deriving the RSBP-NDVI relationships. In this study, we develop a fusion framework to obtain more
accurate high-resolution estimates of precipitation in mountainous areas based on the relationship between
precipitation and vegetation response. More specifically, in addition to RSBP, gauge measurements and different
vegetation datasets will be used in this study to overcome the aforementioned limitations in current
RSBP-NDVI-based schemes. The paper is organized as follows: section 2 describes the development of the fusion
framework; section 3 documents the study area and related datasets; section 4 presents the results of the fusion
framework and discusses impacts of different determinants on the performance of fusion framework; and section
5 summarizes this work.

**2 Framework development**
The satellite-gauge-vegetation fusion framework (Fig. 1) involves three stages of development: 1) vegetation
data merging, 2) precipitation-vegetation regression, and 3) RSBP product remapping, whose details are
described in the following subsections.


**2.1 vegetation data merging**
Vegetation closely interacts with soil moisture and is recognized as a good proxy of precipitation. The remote
sensing technique provides us with various high-resolution vegetation products such as NDVI, EVI (enhanced



vegetation index), LAI (leaf area index), etc. Among the vegetation indices, NDVI, an indicator of plant density
and growth, is chosen as the proxy of precipitation in this study due to its wide availability. Considering the
crucial role of NDVI in deriving precipitation estimates under our framework, we conduct an inter-comparison
in data accuracy between two NDVI datasets (termed as datasets A and B hereinafter) to reduce the error. First,
the systematic errors of both datasets are eliminated by multiplying reduction factor or using simple regression
model. After the correction, the final dataset is then obtained by selecting better element between A and B if the
quality criteria is satisfied otherwise filling an anomaly value.

It should be noted that since the vegetation growth is suppressed or promoted on some land covers (e.g. rivers,
lakes, snow and ice, and urban areas), the vegetation data of these land covers are excluded by filling anomaly
values. Besides, due to the strong influence of farming activities (e.g. irrigation, fertilization, and harvest) on the
crop growth, vegetation data of farmland are excluded as well. We note that although Moran's Index (Li et al.,
2007) is widely employed to detect anomalies in vegetation data (Jia et al., 2011; Duan et al., 2013), it is not used
in this study for its inapplicability in large areas with continuous anomaly pixels (e.g. farmland). As such, we
identify anomaly pixels simply by landuse type: pixels categorized as water, wetland, urban, cropland, snow/ice,
and barren will be identified as anomalies. The detected anomaly pixels are excluded from the original NDVI
dataset and then filled with interpolated values using IDW method so as to generate an optimized NDVI dataset.

Based on the optimized NDVI dataset, the NDVI data at the gauge locations are retrieved with neighbor-average
method (i.e. the value of a certain grid is determined as the average of all its eight neighbors) and will be used for
the precipitation-vegetation regression.

**2.2 precipitation-vegetation regression**
As far as we know, there is no widely accepted form for the precipitation-vegetation relationship. Therefore, the
final regression form will be determined from several candidate relationships, including polynomial, exponential,
logarithmic and linear forms, according to the five metrics: correlation coefficient (R), coefficient of
determination ($R^2$), root-mean-square error (RMSE), mean relative error (MRE) and mean absolute relative error
(MARE), which are given as follows:





$$R = \frac{\sum_{i=1}^{n}(P_i - \bar{P})(O_i - \bar{O})}{\sqrt{\sum_{i=1}^{n}(P_i - \bar{P})^2}\sqrt{\sum_{i=1}^{n}(O_i - \bar{O})^2}} \qquad (1)$$

$$R^2 = \frac{\sum_{i=1}^{n}(P_i - O_i)^2}{\sqrt{\sum_{i=1}^{n}(O_i - \bar{O})^2}} \qquad (2)$$

$$RMSE = \sqrt{\frac{\sum_{i=1}^{n}(P_i - O_i)^2}{n}} \qquad (3)$$

$$MRE = \frac{1}{n}\sum_{i=1}^{n}(P_i - O_i) \qquad (4)$$

$$MARE = \frac{1}{n}\sum_{i=1}^{n}\frac{|P_i - O_i|}{O_i} \qquad (5)$$

where $\bar{O}$ is the mean annual precipitation of all gauges, $O_i$ the mean annual precipitation of gauge $i$, $P_i$ the
estimated precipitation at gauge $i$, and $n$ the total number of gauges.

Also, considering the annual variability of precipitation, the regression model is further determined for two
temporal scales: 1) entire period covering all the study years and 2) individual year of the entire study period. The
**R**egression **M**odels for **E**ntire study period and for **I**ndividual years are thus termed as **RME** and **RMI**,
respectively. RME can utilize the full knowledge of precipitation characteristics of the entire study period,
whereas RMI implies the inter-annual variability. Besides, RME can reasonably reconstruct the precipitation
series of the years when data gaps exist.

The calibration-validation procedure for each candidate model is conducted under three scenarios with different
numbers of gauge and/or years:
a) Fully random: random number of gauges and random number of years are independently used for

calibration and validation;

b) All gauges, partial period: all the gauges will be involved in both procedures, but only 2/3 of years will

be randomly chosen for calibration and the other years for validation;

c) Partial gauges, entire period: all years will be used, but only 1/3 of gauges will be randomly chosen for

calibration and other gauges for validation.

For each scenario, the calibration-validation procedure will be performed for one hundred samples determined





based on the above criteria and the six evaluation metrics (i.e. R, $R^2$, $E_{RMS}$, $E_{MA}$ and $E_{MAR}$) will be calculated for
each sample accordingly. The best model is then determined based the metrics.

**2.3 RSBP product remapping**
With the optimized vegetation dataset and precipitation-vegetation regression model, the RSBP product is then
remapped over the study region. Thanks to the finer resolution of NDVI dataset than RSBP product and the
accurate estimate of precipitation by gauges, the remapped RSBP product is expected to provide more detailed
spatial characteristics of precipitation over mountainous areas.
**3 Study area and datasets for framework application**
**3.1 Study area**
The Nu-Salween basin (Fig. 2), where 6 million people are living, is one of the largest river basins in South Asia
and spreads across three countries with an area of 324,000 $km^2$. This study focuses on the Chinese part of the
Nu-Salween basin (termed as Nu river basin hereafter), where the elevation ranges from 446 m to 6134 m and the
narrowest part is only 24 km. The annual precipitation of the Nu river basin ranges from 400 mm to 2000 mm with
an average of 900 mm and the mean annual runoff is 69 $km^3$. The precipitation of the Nu river basin generally
decreases from southwest to northeast and demonstrates high variability due to mountain weather systems (e.g.
the difference in annual precipitation between the mountaintop and valley of Gongshan is larger than 1000 mm).
Thanks to the adequate rainfall and minimal human perturbation, the Nu river basin has an extensive vegetation
coverage with the dominant type as grassland in the Qinghai-Tibetan Plateau (upper basin) and mixed forest in
Yunnan province (lower basin). However, the dense vegetation cover increases the difficulty in conducting
precipitation observations and only 13 gauges are very unevenly distributed over the whole basin of 142,479 $km^2$,
which makes it highly challenging to obtain the accurate spatial precipitation characteristics with traditional
interpolation approaches. Although the RSBP products are available for this area, they are too course (usually with
a spatial resolution of ~50 km) to capture the high spatial variability of precipitation.


Considering the limited number of gauges (i.e. 13) in the Nu river basin, an enlarged area covering 23°N–33°N




and 91°E–101°E is chosen for the application of the fusion framework, where 59 gauges are available and the
climatic and topographic conditions are consistent with the Nu river basin.

**3.2 Datasets**
**3.2.1 Vegetation data**
In this study, we use the two MODIS (moderate resolution imaging spectoradiometer) vegetation products,
MOD13A3 (termed MOD hereafter) and MYD13A3 (termed MYD hereafter), in the application of the fusion
framework. Both the MOD and MYD datasets contain 10 sub-datasets consisting of NDVI, EVI and pixel
reliability. The pixel reliability is an accuracy metric of the data quality pixel and has four valid values: 0 for good
accuracy, 1 for marginal accuracy, 2 for snow/ice, and 3 for cloud. Based on the pixel reliability information, the
NDVI values are either selected for corresponding pixel reliability levels being 0 and 1 or discarded as anomalies
otherwise.

The MOD dataset is used as benchmark while MYD is taken as the alternative for occasions when MOD data are
missing or have large uncertainties. Since both the MOD and MYD datasets are extracted from different satellites
at different transit times, systematic errors may exist in the difference between the two datasets. As such, we
construct two regressions to remove their systematic errors: one is based on a subset with both MOD and MYD of
good accuracy (reliability = 0), and the other on a subset with MOD of marginal accuracy (reliability = 1) and
MOD of good accuracy (reliability = 0). After the removal of systematic errors, a **m**erged dataset of MOD and
MYD (termed MMD hereafter) is generated under the criteria given as follows:

$$MMD = \begin{cases} MOD & (MOD == 0) \\ MYD & (MOD > 1 \ \& \ MYD == 0) \\ MOD & (MOD == 1 \ \& \ MYD == 1) \\ NULL & (MOD > 1 \ \& MYD > 0) \end{cases} \qquad (6)$$

The annual MMD dataset is then calculated by averaging the 12 monthly images.

**3.2.2 Landuse data**
The landuse dataset MCD12Q1 Version 51 (MODIS/Terra+Aqua Land Cover Type Yearly L3 Global 500m SIN
Grid V051) in period of 2001-2013 is used to identify the outliers of MMD and the IGBP (International Geosphere





Biosphere Programme) classification is adopted in this study for its wide applications. Due to mismatch in spatial
resolutions between MMD and MCD12Q1 datasets, the MCD12Q1 dataset is upscaled to 1km as MMD for outlier
identification. It should be noted that for any of the four 500 m pixels in MCD12Q1 classified as water, urban,
snow or ice, the upscaled 1 km pixel will be assigned with a missing value (i.e. -9999) and the corresponding
NDVI pixel will be identified as an outlier.

**3.2.3 Weather data**
Datasets consisting of daily precipitation and air temperature collected at the 59 gauges in the study area are
obtained via the China Meteorological Data Sharing Service system (cdc.nmic.cn). The air temperature
measurements will be used for dependence analysis later in Section 4.5.

**4 Results and discussion**
**4.1 Model calibration and validation**
Based on the results of six evaluation metrics for different regression form candidates (not shown here), the
$2^{nd}$-order polynomial is chosen as the regression model form in this study:

$$p = a \times NDVI^2 + b \times NDVI + c \tag{7}$$

where $p$ denotes precipitation amount in mm, and $a$, $b$ and $c$ are regression coefficients. The results of regression
coefficients and evaluation metrics are given in Table 1, and the NDVI-precipitation relationships for the study
period are demonstrated in Fig. 3.

The best performance of the regression model is found within $0.2 < NDVI < 0.7$ and 400 mm year$^{-1}$ < p < 1500 mm
year$^{-1}$. Larger errors are found at pixels with NDVI larger than 0.7 or annual rainfall larger than 1500 mm,
implying the water supply is no longer a determinant of vegetation growth as annual rainfall exceeds a certain
threshold.

In general, the RMIs demonstrate better performance than RME, which can be attributable to the less variability of
precipitation in a single year than the whole study period. It is also noted that the $R^2$ values of RMIs for drier years





(2003, 2009 and 2011) are less than wetter years, indicating the weaker coupling effect between vegetation growth
and precipitation.

The performance of regression models is assessed under three scenarios as described in Section 2.2. A total of 300
tests are conducted and performance metrics (i.e., R, $R^2$, RMSE, and MARE) are calculated accordingly (Fig. 4
and Table 2). The high R values (> 0.85) indicate a strong correlation between NDVI and precipitation
independent of sampling method. Also, the regression models demonstrate good performance with $R^2$ larger than
0.75 and MARE less than 20%. In addition, the metrics of regression models fluctuate around that of the RME
with narrow inter-quartile ranges, indicating the regression models have remarkable consistency with the RME
model.

Scenario a is designed to examine inter-annual stability in the performance of   regression models, where the good
performance indicates the acceptable ability of the RME model in estimating precipitation during periods when
precipitation measurements are not available. Scenario b and c investigate the impacts of spatial and temporal
coverages of measurements, respectively. It is noteworthy that under scenario b better performance in regression
models is observed as compared with scenario c, implying greater importance of spatial coverage of
measurements in conducting the regressions. In addition, the results of calibration is better than validation as
revealed by all metrics criterions as expected. However, the differences between calibration and validation are not
significant, implying the consistent performance of regression models under various scenarios.

The performance of RME is further assessed by comparing the estimates against observations (Fig. 5), and good
agreement between estimates and observations is observed. It should be noted the RME shows difficulty in
estimating precipitation larger 2000 mm ( cf. the dashed line in Fig. 5), implying the limitation of the fusion
framework inherited from the oversaturation effect of vegetation index.

**4.2 Spatial characteristics of precipitation**
The spatial characteristics of precipitation of the study area are investigated with RME for the whole study period
(Fig. 6). The annual precipitation is observed to decrease from south to north and from west to east and its spatial
variability is clearly demonstrated. Two "hot-spot" regions, whose annual precipitation exceeds 1500 mm, can be





identified in the study areas: one near south border and the other close to southwestern mountain border. The east
part of the Nu river basin featuring a dry and warm climate receives an average annual precipitation of 800 mm
with large inter-annual variability.

The residuals (differences between observed rain and estimated rain) represent the part of the precipitation that
cannot be explained by NDVI alone and the residual map (Fig. 7) is produced by interpolating gauge residuals to
study area using IDW method.

**4.3 Model performance comparison**
The performance between IDW approach, TRMM product and the fusion framework is compared in this section.
In general, the IDW approach is unable to demonstrate the high spatial variability though it can capture the general
spatial distribution of whole basin (Fig. 8a) as TRMM (Fig. 8b). Due to the coarse spatial resolution, TRMM
cannot capture the high variability in the river valley where the elevation varies significantly.

To demonstrate the advantage of the fusion framework, a cross-validation is conducted against the randomly
sampled gauge observations by varying the number of samples (1 - 40). The cross-validation shows higher RMSE
for the IDW approach, followed by TMMM and RME (Fig. 9a). A higher mean MRE of 15% is observed for
TRMM than IDW (8%) and RME (5%) while the difference in MARE are minimal between TRMM and IDW.
The results indicate an overestimated precipitation by TRMM as compared to gauge observations. Table 3
summarized the maximum, minimum and mean values of each method and shows the relative difference
between RME and other two methods. On average, RMSE of RME is smaller than that of IDW and TRMM by
20.4% and 17.4%, respectively. In general, the fusion framework demonstrates better performance than the other
approaches.

To further evaluate the performance of RME, the annual averages of precipitation of five hydrological stations
(locations refer to Fig. 6) and whole basin estimated by the three approaches (IDW, RME and TRMM) are
compared (Fig. 10). At the whole basin scale, the estimate by RME is 5.2% higher than that of IDW while 7.9%
lower than TRMM. Although the difference between the three approaches is minimal at the basin scale, the
difference at the sub-basin scale is remarkable. In the upstream region (i.e., Gongshan sub-basin) located in




Tibet Plateau, TRMM overestimates precipitation by 13.2% while IDW underestimates by 7.6% as compared
with RME. In the other four downstream sub-basins, estimates by RME are larger than those by IDW and TRMM.
In general, in the midstream and downstream regions with large variability in terrain height, RME gives larger
estimates of precipitation than IDW and TRMM.

**4.4 influence of different vegetation index**
Considering the possible degradation in model performance caused by oversaturation of NDVI in high biomass
areas, another vegetation indicatwor, Enhanced Vegetation Index (EVI), is suggested as an alternative for
estimating vegetation growth (Matsushita et al., 2007; Liao et al., 2015). As such, we also test the fusion
framework with EVI in addition to NDVI and the results are assessed against the gauge observations.

Based on the chosen metrics, EVI is found to outperform NDVI with better regression quality (Table 4):
EVI-based regression model gives higher $R^2$, smaller RMSE and MARE compared to the NDVI-based model.
Also, remarkable difference is observed in the precipitation estimates based on the two vegetation indices (Fig.
11). It is noted that the curvature of EVI-based model is larger than NDVI-based model, suggesting higher
sensitivity of EVI-based model in humid environment. Although the EVI-based model demonstrates better
performance than the NDVI-based one, it should be noted that NDVI is the most popular vegetation index used in
operational applications among the available vegetation indices. Besides, NDVI has a relative longer temporal
coverage compared to other vegetation indices. For instance, the AVHRR (Advanced Very High Resolution
Radiometer) NDVI data are available since 1982 with the coverage of entire Earth. As such, under scenarios when
EVI is unavailable, NDVI is a satisfactory index that can be used in the fusion framework.

**4.5 Influence of other ambient determinants**
One major assumption of the proposed framework is that precipitation is the only determinant of vegetation
growth and thus NDVI is regarded as a proxy for precipitation. However, other ambient factors, such as soil
properties, solar radiation, air temperature, elevation, etc., may significantly influence the vegetation growth as
well as NDVI values. Considering the data availability of various ambient factors, air temperature and elevation,
in addition to NDVI, are adopted as extra determinants to establish the regression models, which are thus termed





as RME+T and RME+H for air temperature and elevation, respectively. We note that for simplicity, the extra
determinants are assumed to have linear relationship with precipitation.

The difference in $R^2$, RMSE, and MARE between the three models are minimal and the regression coefficients of
the three models are very close to each other (Table 5). The negative regression coefficient of temperature in
RME+T indicates that precipitation decreases as the temperature increases, whereas the negative regression
coefficient of elevation in RME+H indicates that precipitation decreases as the elevation increases. Since the
temperature decreases with the increase in elevation, RME+T and RME+H essentially provides consistent
estimates of precipitation. It is also noted the added information by extra determinants (i.e., air temperature and
elevation) is in fact minimal. As such, we consider RME only based on vegetation index as a simple and efficient
model for precipitation estimation.

**5 Conclusion**
In this study, a satellite-gauge-vegetation fusion framework has been developed for estimating the precipitation in
mountainous areas by establishing regression relationship between gauge-based precipitation observations and
satellite-based vegetation dataset. The fusion framework was then applied in the Nu River basin of Southwest
China for estimating precipitation between 2001 and 2012.

The fusion framework for the Nu River basin adopted a 2nd order polynomial form and demonstrated promising
ability in capturing the high spatial variability of precipitation in the river valley. Six evaluation metrics, including
R, $R^2$, RMSE, MRE and MARE, indicated good performance of the fusion framework in precipitation estimation.
The performance of the fusion framework was also compared with the IDW approach and TRMM product and the
comparison results indicated that the fusion framework generally outperformed other approaches in estimating
precipitation in mountainous areas according to the chosen metrics. On average, the RMSE of the fusion
framework is 20.4%, 17.4% smaller than that of IDW and TRMM, respectively. MRE of the fusion framework is
1.2%, 71.5% smaller than that of IDW and TRMM. MARE the fusion framework is 18.9%, 28.3% smaller than
that of IDW and TRMM.



The success of application of the fusion framework in the Nu River sheds light on the precipitation estimation in
mountainous areas by using multi-source datasets. One limitation of this work that should be appreciated is the
limited application in a single mountainous basin. Also, possibilities of using other vegetation datasets under this
fusion framework can be explored in the future.





*Acknowledgments.* The study is supported by NSFC under grant U1202231 and 51409147, National Key
Technology Support Program under grant 2011BAC09B07-3 and by China Postdoctoral Science Foundation
under grant 2015T80093. The authors thank China Meteorological Administration, MODIS NDVI, Tropical
Rainfall Measuring Mission (TRMM) and the Shuttle Radar Topography Mission (SRTM) for providing the data
used in this study.

**Appendices**
**Merging of NDVI datasets**
The merge work improved the accuracy as expectation, Fig. A1 shows monthly error rate of MOD and MMD.
Error rate is defined as the ratio of the pixel which quality value is over 1. From Fig. A1, the error rate of MMD
decrease reduced every month. For average over 5% error rate is reduced by the merge work. Some months over
20% error rate is reduced.

Fig.A2 shows that the accuracy of MMD is significantly improved in a ridge area covering about 23°10′ N–
23°40′ N and 98°30′ E–99°E. Fig. A2b shows NDVI value near right and left boundary is underestimated by
MOD. Fig.A2c shows NDVI value middle boundary is underestimated by MYD. Both the underestimation near
the boundary of MOD and MYD is amended (Fig. A2a). Fig.A3 show the three NDVI series for one rain gauge.
Comparing with MOD series, MMD products improved accuracy mainly at wet season of May to October. In
these months the NDVI values are easy to be underestimated due to the cloudy weather.

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






**Table 1** Regression model performance and regression coefficients.

| Year | Mean (mm) | $R^2$ | RMSE (mm) | MARE (%) | a | b | c |
|---|---|---|---|---|---|---|---|
| 2001 | 961 | 0.91 | 138 | 10.6 | 3038.1 | -345.3 | 359.8 |
| 2002 | 887 | 0.90 | 119 | 10.2 | 1354.7 | 687.5 | 212.0 |
| 2003 | 828 | 0.75 | 155 | 14.0 | 1700.2 | -115.5 | 472.7 |
| 2004 | 1018 | 0.89 | 171 | 12.4 | 3784.3 | -1047.7 | 517.4 |
| 2005 | 810 | 0.93 | 97 | 9.5 | 2465.4 | -265.0 | 363.2 |
| 2006 | 737 | 0.88 | 122 | 11.4 | 2065.2 | -112.2 | 287.5 |
| 2007 | 928 | 0.84 | 184 | 14.6 | 2306.9 | 53.5 | 286.4 |
| 2008 | 960 | 0.91 | 121 | 9.4 | 2504.0 | -258.1 | 433.5 |
| 2009 | 726 | 0.89 | 119 | 13.2 | 2091.3 | -168.0 | 294.5 |
| 2010 | 937 | 0.94 | 124 | 9.1 | 4094.8 | -1293.3 | 512.6 |
| 2011 | 824 | 0.84 | 167 | 14.2 | 4697.8 | -2613.7 | 792.7 |
| 2012 | 791 | 0.89 | 114 | 10.6 | 1966.4 | 3.5 | 308.1 |
| RME | 848 | 0.83 | 174 | 15.2 | 2670.4 | -471.2 | 409.2 |



**Table 2** Statistics of regression models for validation and calibration under three scenarios.

| Scenario | Statistics | Calibration | | | | Validation | | |
|---|---|---|---|---|---|---|---|---|
| | | R | $R^2$ | RMSE (mm) | MARE (%) | R | RMSE (mm) | MARE (%) |
| a | mean | 0.91 | 0.83 | 175 | 16.6 | 0.91 | 173.9 | 16.8 |
| | max | 0.92 | 0.85 | 186.2 | 17.8 | 0.94 | 211.8 | 19.9 |
| | min | 0.9 | 0.81 | 161.1 | 15.7 | 0.88 | 141 | 13.2 |
| b | mean | 0.92 | 0.84 | 166.6 | 15.8 | 0.91 | 186.1 | 17.8 |
| | max | 0.94 | 0.89 | 207 | 19.7 | 0.95 | 229.7 | 23.3 |
| | min | 0.89 | 0.8 | 126.2 | 12.8 | 0.89 | 148.6 | 12.9 |
| c | mean | 0.91 | 0.82 | 172.7 | 16.5 | 0.91 | 180.8 | 17.3 |
| | max | 0.95 | 0.91 | 207.9 | 19.1 | 0.94 | 204.8 | 24.4 |
| | min | 0.85 | 0.73 | 144.6 | 13.9 | 0.85 | 143.4 | 13.9 |












**Table 3** Performance comparison between IDW, RME and TRMM

| Method | Statistics | RMSE (mm) | MRE | MARE |
|---|---|---|---|---|
| | max | 273 | 0.1 | 0.26 |
| IDW | min | 249 | 0.08 | 0.23 |
| | mean | 223 | 0.05 | 0.21 |
| | max | 220 | 0.17 | 0.24 |
| TRMM | min | 213 | 0.16 | 0.23 |
| | mean | 203 | 0.15 | 0.22 |
| | max | 183 | 0.07 | 0.18 |
| RME | min | 177 | 0.05 | 0.17 |
| | mean | 168 | 0.04 | 0.16 |
| RME-IDW | max | -32.9 | -33 | -30.5 |
| | min | -26.3 | -9.8 | -21.4 |
| (%) | mean | -20.4 | -1.2 | -18.9 |
| RME-TRMM | max | -16.8 | -59.5 | -23.8 |
| | min | -16.6 | -66 | -25.9 |
| (%) | mean | -17.4 | -71.5 | -28.3 |


**Table 4** Regression model performance and coefficients of regression

| | $R^2$ | RMSE (mm) | MARE (%) | $a$ | $b$ | $c$ |
|---|---|---|---|---|---|---|
| NDVI | 0.83 | 174.7 | 14.8 | 2670.4 | -471.2 | 409.2 |
| EVI | 0.87 | 143.8 | 12.4 | 5129.6 | 702.5 | 254.7 |



**Table 5** Results of two regression models established with extra independent variables: RME+T for temperature,
RME+H for elevation

| Model | $R^2$ | RMSE (mm) | MARE (%) | $a$ | $b$ | $c$ | Extra $b$ |
|---|---|---|---|---|---|---|---|
| RME | 0.83 | 174.7 | 15 | 2670.4 | -471.2 | 409.2 | -- |
| RME+T | 0.84 | 172.6 | 15 | 2728.8 | -496 | 407.3 | -0.2 |
| RME+H | 0.84 | 172.6 | 15 | 2838.4 | -638.7 | 492.9 | -0.02 |









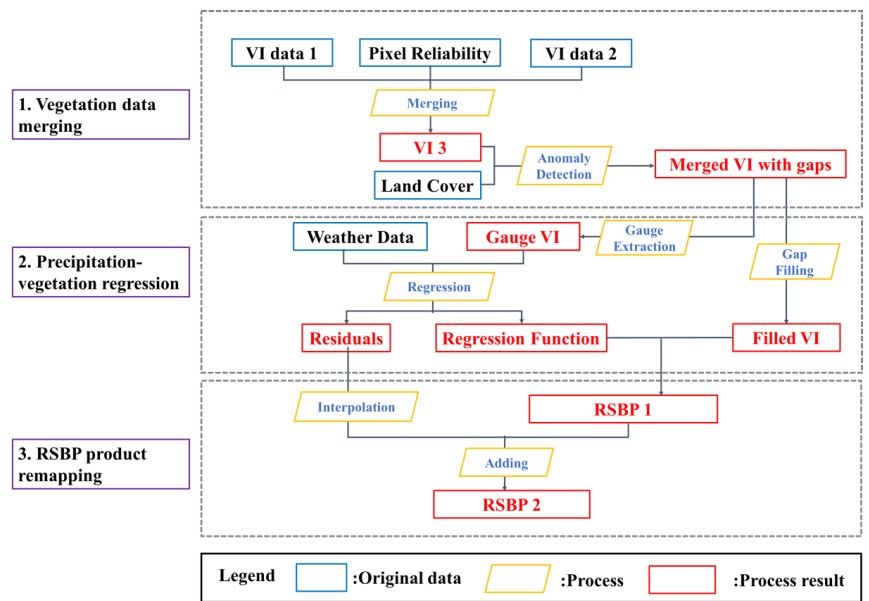


**Figure 1** Flow chart of the satellite-gauge-vegetation fusion framework development.

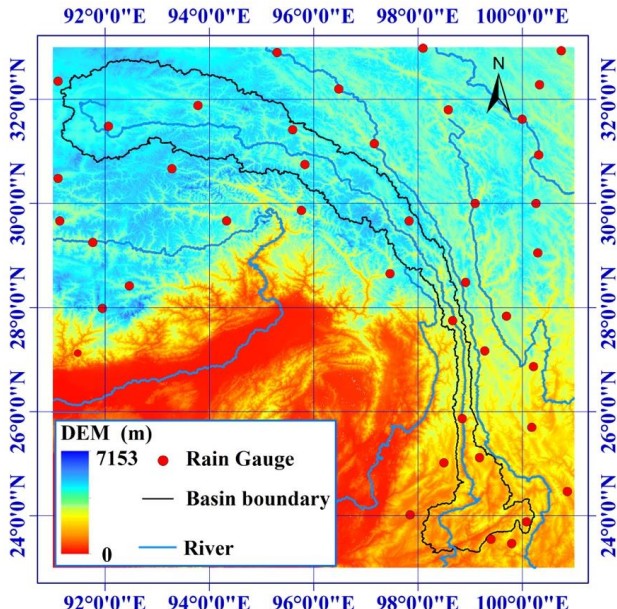


**Figure 2** Terrain map of the study area (the Nu-Salween basin and its adjacent areas). The red dots denote the
weather stations used in this study.






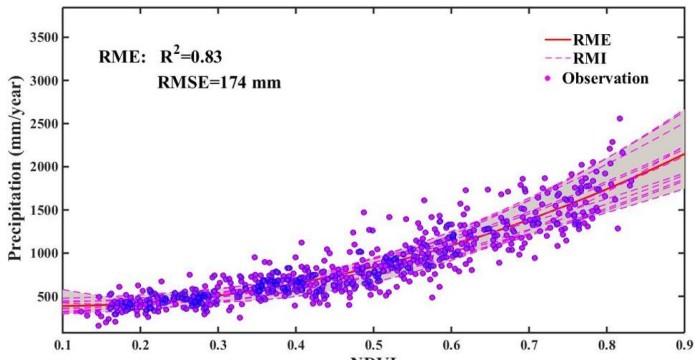


**Figure 3** The NDVI-precipitation relationships.


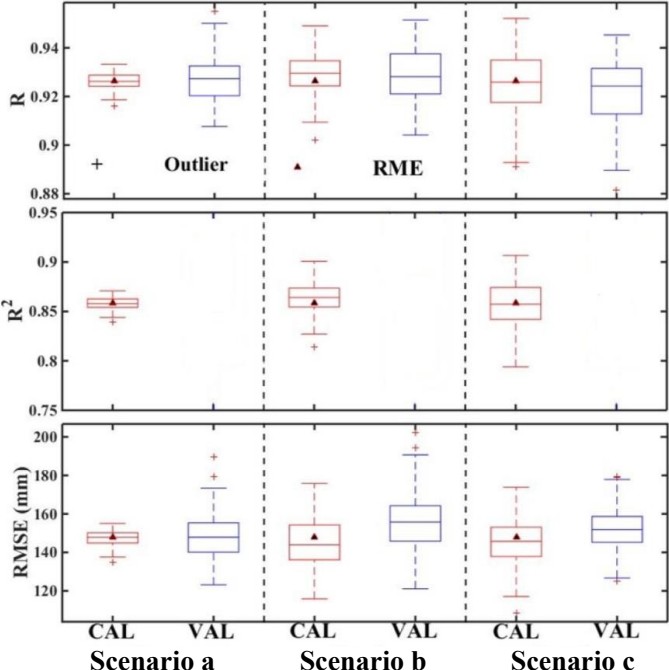


**Figure 4** Box plots of R, $R^2$, RMSE of RME model under three scenarios: a) fully random; b) all gauges, partial
period; and c) partial gauges, entire period. Details of the three scenarios refer to Section 2.2.





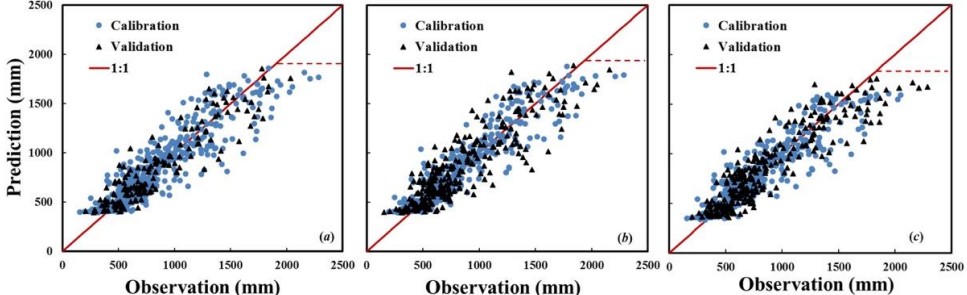


**Figure 5** Comparison in annual precipitation between the gauged measurements and predictions by the regression

model for scenario a) fully random; b) all gauges, partial period; and c) partial gauges, entire period. Details of the

three scenarios refer to Section 2.2.


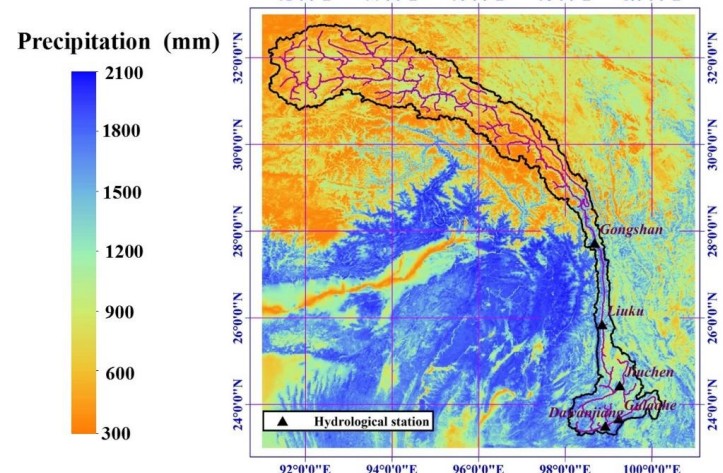


**Figure 6** Average annual precipitation distribution of 2003-2012 from RME.







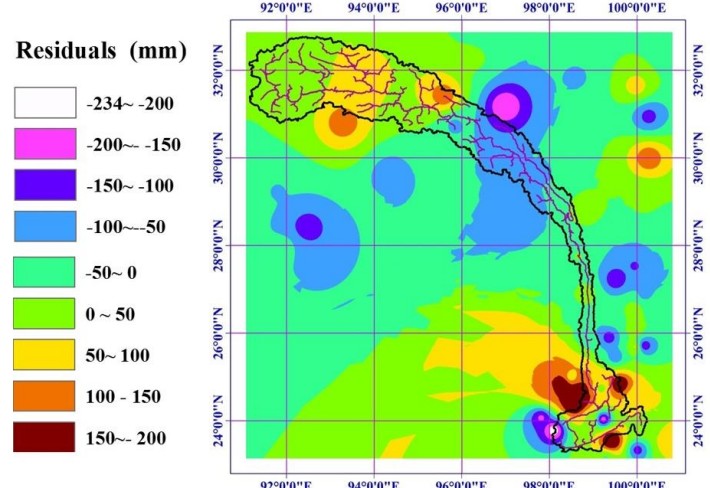


**Figure 7** Average residuals distribution of 2003-2012 from RME.



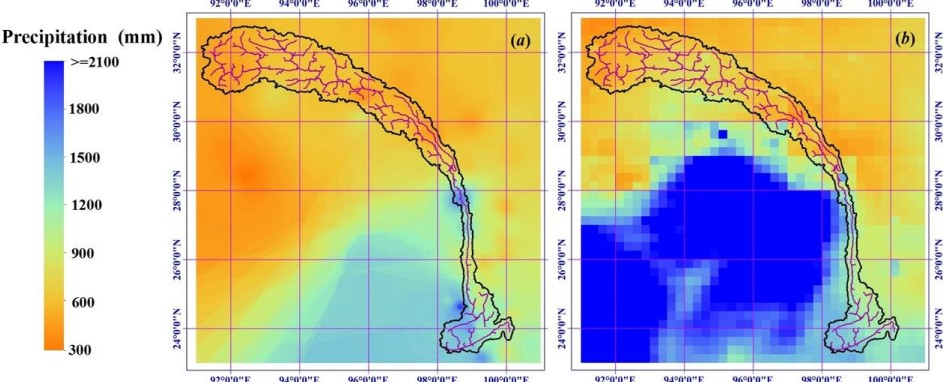


**Figure 8** spatial distribution of mean annual precipitation of 2003-2012 estimated by (a) IDW and (b) TRMM.












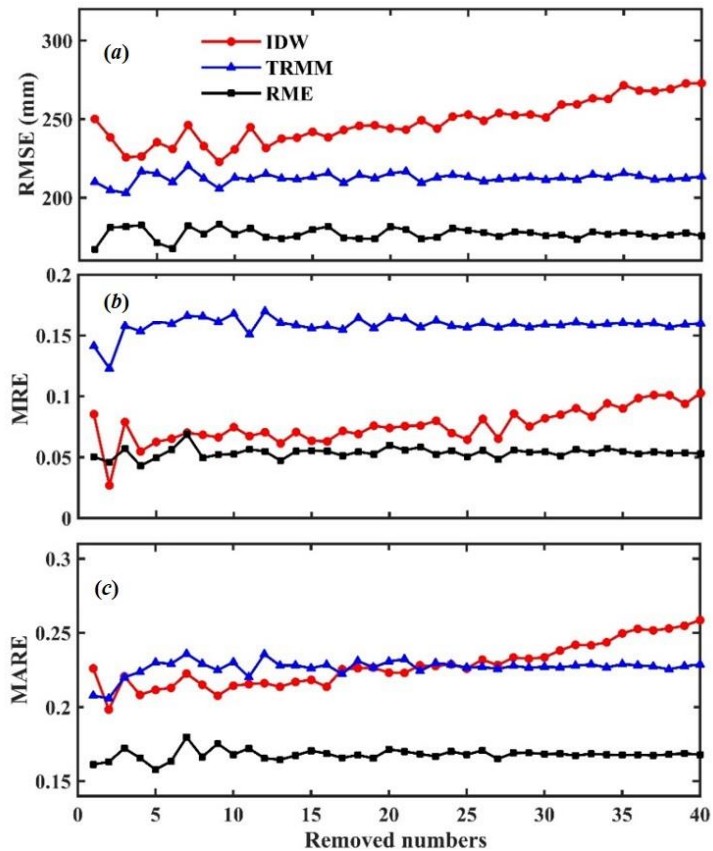


**Figure 9** Performance of RMSE, MRE and MARE for three methods in different remove numbers.

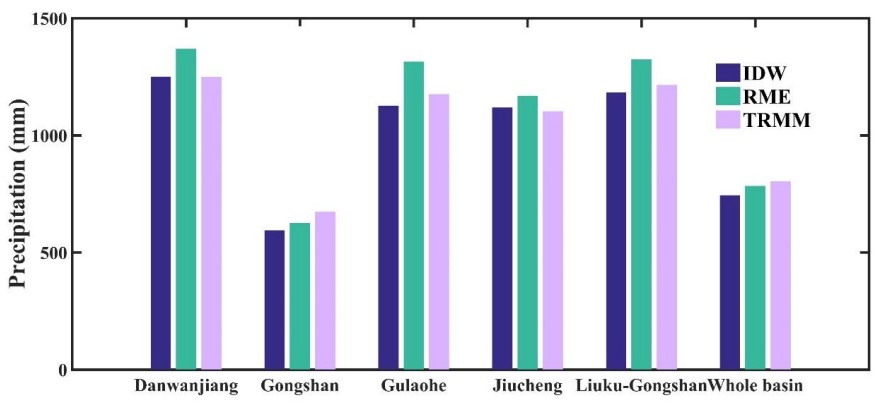


**Figure 10** Comparison of average annual precipitation at sub-basin scale got from IDW, RME and TRMM

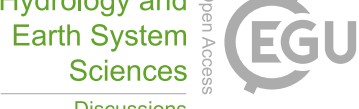




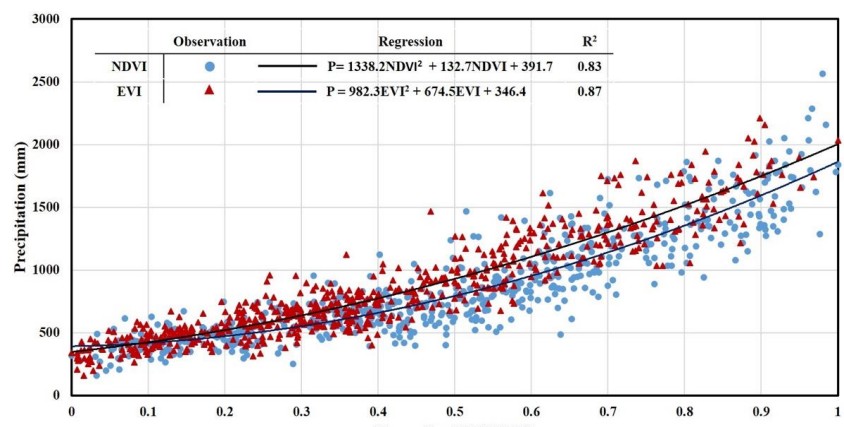


**Figure 11** Regression relationship between annual precipitation and normalized NDVI/EVI


**Figure A1** Monthly Error rate of MOD, MYD and MMD






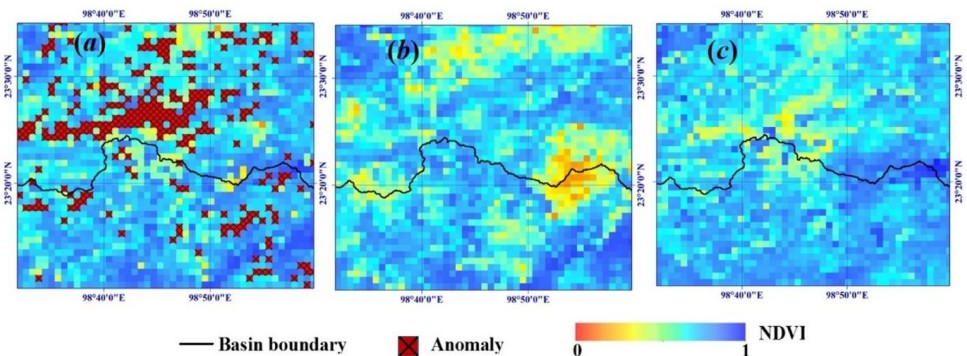


**Figure A2** Comparison of three NDVI products over a ridge area on June 2006, (*a*) for MMD, (*b*) for MOD, (*c*)
for MYD





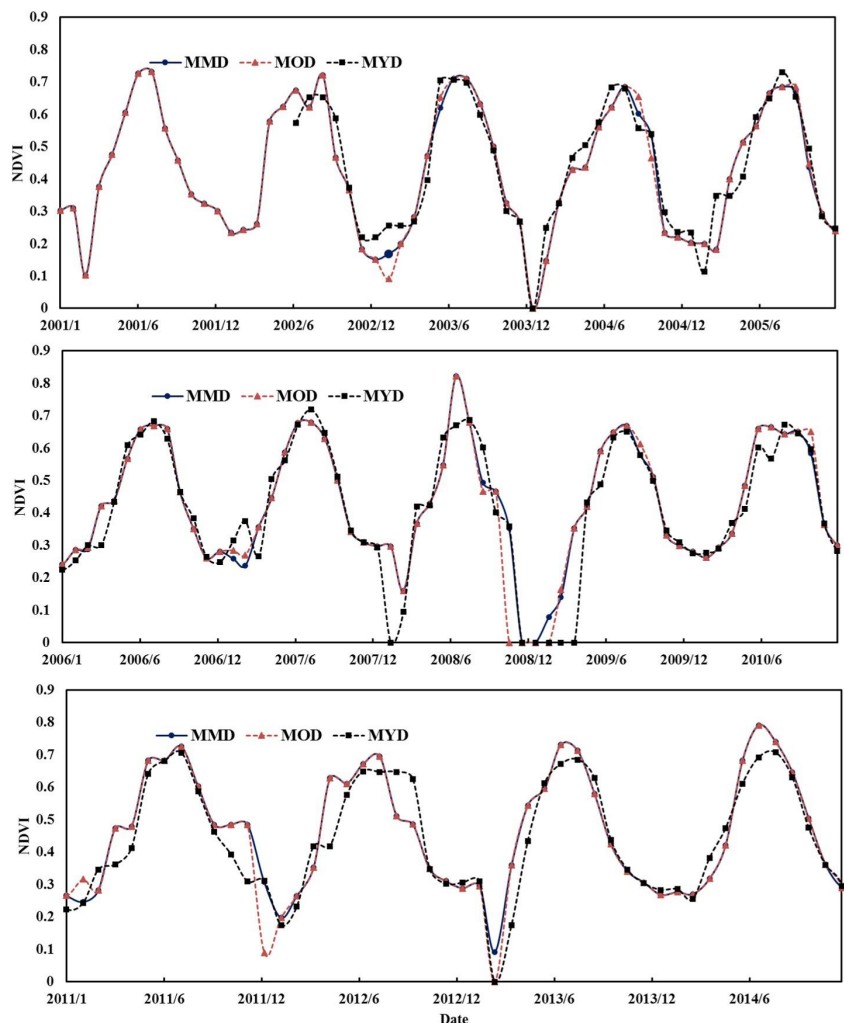


**Figure A3** Comparison of three NDVI monthly times series over one gauge
