# Peer review of "Remapping annual precipitation in mountainous area based 1 on vegetation pattern: a case study in the Nu River basin 2 Xing Zhou1, Guang-Heng Ni1, Chen Shen1, Ting Sun1 3 4 5 1) State Key Laboratory of Hydro-Science and Engineering, Department of Hydraulic"

_Hydrology and Earth System Sciences, 2016_

## Referee Comment (RC1) · Anonymous Referee #1 · 7 Dec 2016

Overall: Data fusion across satellite and gauge precipitation data has been widely concerned around the world in the last decade and also has been investigated by many studies in China. In this study, the authors aim to develop a fusion framework to improve the precipitation estimation in mountainous areas. The framework is then applied to the Nu river basin, a place with high altitude and complex topography and also with distinctive climate. From this perspective this study is meaningful. I would suggest that this paper be accepted with a few minor revisions.

1. Page-7, line 158, "59 gauges are available", which are not consistent with the number of stations in Figure 2.

2. Page-7, line 164, it is suggested that a sentence be given to describe the temporal and spatial resolution of MOD13A3 and MYD13A3 vegetation products.

[Figure]

3. Page-8-9, line 208-209, " It is also noted that R2 values of RMIs for drier years are less than wetter years......". Time-lag effects of vegetation responses to precipitation can be considered to explain this phenomenon.

4. Page-12, line 299, "the negative regression coefficient of temperature in RME+T indicates that precipitation decrease as the temperature increase". I don't agree that your explanation of negative regression coefficient of temperature shows that that precipitation decreases as the temperature increase.

5. Some detailed should be paid more attention. For example, line 191, correct website should be given; In Figure 2, legend of DEM should not be 0 in this region.

---

## Referee Comment (RC2) · Anonymous Referee #2 · 20 Dec 2016

This is a very interesting and clearly written paper on precipitation distribution in a mountainous river basin. Mountains, in particular high mountains in Asia, are the source of freshwater for many people, but the observation of precipitation is far from enough. For example, the first item of Scientific Questions of GEWEX under WCRP is the observation of precipitation. As such, the topic of this paper is within the scope of HESS, and is a hot topic in hydrology. As already described above, the paper is very clearly written, thus easy to follow. Therefore, I basically want to recommend this paper toward publication in HESS. Nevertheless, I would also mention that there are concerns, comments, or suggestions as written below.

1. In the title, abstract and also in any other part of this paper, the authors should clearly mention that the target of this research is annual mean or climatological mean precipitation amount. The authors compared their product with IDW (of gauge-based

precipitation) and a TRMM-based product. I can accept this product is better for annual mean and climatological mean. However, IDW and TRMM can provide us short time-scale data, such as daily data. This is a major difference between these products against the product which the authors try to show in this paper.

2. It is also recommended to clearly describe that this study is a case study for a specific area, in the title, abstract, and also in other places of this paper. Why I recommend so is described in the following. As shown in Figure 2 and Figure 6, in this target area, precipitation amount is larger for areas of lower elevation. In areas of lower elevation, it is usually expected that air temperature is warmer. It means, in the areas of lower elevation in this study, both precipitation and temperature are better, favorable, for NDVI. I suspect, this the reason why the authors can get a clear positive relationship between precipitation and NDVI. But, this is specific to this particular area. Of course, there could be other similar areas in the world. However, there must also be areas with different characteristics between NDVI and precipitation. Thus, the authors are requested to explicitly describe in the title and the abstract and also in relevant places of the main text that this study is a case study with such characteristics shown in Figures 2 and 6.

3. This comment is a comment following the above comment. Very in general, precipitation is larger if elevation is higher. This is in contrast with what is seen in Figures 2 and 6. But, I need to add to "precipitation is larger if elevation is higher". Such a general tendency is probably true to 2000m or 3000m in elevation. In this regard, "we note that for simplicity, the extra determinants are assumed to have linear relationship with precipitation" is somewhat suspicious because this area has elevation up to 7000m. Is it possible to make a figure in which horizontal axis is elevation and vertical axis is precipitation (and NDVI) using observed precipitation data and NDVI data. There might be a positive relationship between elevation and precipitation up to 2000 or 3000m in elevation, and a negative or flat relationship between elevation and precipitation after it. Also, because NDVI favors large precipitation and warmer temperature (= lower elevation), the response of NDVI is different up to 2000 or 3000m and after it. NDVI-

precipitation relation may depend on elevation bands such as lower than 3000m and higher than 3000m (of course, I do not know it would be 3000m or not which changes the relationship), but such an analysis was not done in this study as far as I can see. I think the authors can easily analyze.

4. Although target temporal and spatial scales are different, a recent study submitted to HESSD (Beck et al., 2016) provides a globally distributed precipitation data (called MSWEP) in which mountainous precipitation is corrected for gauge under-catch and also orographic effect was introduced by inferring catchment-average P from stream-flow (Q) observations at 13762 stations across the globe. I found mountainous precipitation is somehow well represented in the product by Beck after downloading the data from http://www.gloh2o.org and by making figures of the data by myself. Thus, it is recommended to have discussion in terms of Beck et al. (2016).

Specific minor comments are as follows.

- A map of sub-basins is better to be provided (for Figure 10).

- I do not think Figure 7 is good to show. I can understand if the authors show the difference between Figure 8 and Figure 6. At least, I would say Figure 7 is awkward.

- I also suspect that all the areas in Figures 6, 7, 8 are appropriate to show. I mean, there is almost no observation station in the left lower quarter of Figure 2. Then, I suspect whether the values of precipitation amount shown in Figure 6, 7, 8 for the left lower quarter of those figures are enough valid, particularly for the main product of this study and the IDW-based output.

Very finally, this is probably out of scope of this paper, but I am interested in whether major precipitation season is the same over this particular region. I mean, summer precipitation and winter precipitation (particularly solid precipitation like snow) may have different responses.

---

## Referee Comment (RC3) · Anonymous Referee #3 · 21 Dec 2016

General comments:

Gridded precipitation data are very useful for hydrological application and others and ground-observation-based ones have been developed for many regions. This study investigated a methodology to develop gridded precipitation data for the Nu River basin based on the ground-observed precipitation combined with vegetation indices. As an interpolation method, the data fusion may be a strong tool especially for a sparse observation area. Its application to the Nu River with sparse observations may contribute to expanding a hydrological knowledge. This paper requires some more analyses to make readers more convinced of the effectiveness. Therefore, I recommend to revise this manuscript based on comments below before publication.

Major comments:

[Figure]

The regression for RME uses all the data at the gauge stations and for the entire years. Such regression provide the climatological mean relationship between precipitation and NDVI. This regression cannot be applied for individual year as seen Table 1 where the coefficients distinct vary with year. Authors must mention the limitation of the proposed method in an appropriate paragraph.

Judging from Figure 6, climatological mean annual precipitation seems to depend on elevation. The dependence of precipitation on elevation is a well-known fact. In order to clarify the effectiveness of your method, it is better to compare the geographical distributions of climatological mean precipitation between your methods and a method by a regression between precipitation and elevation. A figure of the difference between the two may provide an important suggestion about strong points of your method.

Specific comments:

L66: The objective of this study should be more specified in terms of spatio-temporal scale: climatological annual mean and 1 km.

L103: The center of a certain box is not used in computing the averaged value of the grid box? If so, please provide a reason.

L157: The number of rain gauge stations in the Nu River basin seems smaller than 13 as long as it is judged from Figure 2.

L158: Moreover, the 59 stations are not plotted on Figure 2.

L159: Please explain what you mean by "climatic and topographic conditions are consistent with the Nu River basin."

L162: You use three terminologies: uncertainty, reliability, and accuracy in the 3.2.1 subsubsection. Please provide short definitions if you distinguish them in this manuscript.

L163: In this paragraph, please provide the spatio-temporal resolution of the MODIS

dataset. L176: "m" in "merged" should be in standard font.

L185: How do you classify an upscaled 1 km pixel when it is composed of two forest and two cropland pixels? You do not consider the mixed pixel? If so, please provide this information.

L196: Readers may know the rationale about the selection of the regression form. Not by "not shown here" but "judging from Figure 3" is better.

L202: Did you use these results when you draw Figure 6? If not, why don't you use these important information?

L220: Two spaces exist between of and regression.

L222: Scenario should be Scenarios.

L242: Please provide rationale about the use of the IDW method here.

L264: Precipitation by the RME method often the largest among the three in Figure 10 is reasonable? Can you validate this precipitation estimates larger than the other two by sub-basin water balance such as observed river discharge = P – E. In other words, large amount of precipitation is better than small one in order to explain the observed river discharges.

L295: As mentioned above, in addition to RME+T and RME+H, how the regression of T or H onto precipitation works for producing climatological mean annual precipitation like Figure 6.

Figures 1: Font color should be the same as in outline color of boxes.

Figures 2, 6, 7, 8, and A2: Minutes and seconds should be removed from the annotations of the coordinates. Font size should be enlarged so as to see them easily.

Figure 4: Please provide explanations about two symbols in the figure caption.

[Figure]

---

## Author Comment (AC1) · 23 Dec 2016

**Responses to Anonymous Referee #1:**

*General Comment:*

*Data fusion across satellite and gauge precipitation data has been widely concerned around the world in the last decade and also has been investigated by many studies in China. In this study, the authors aim to develop a fusion framework to improve the precipitation estimation in mountainous areas. The framework is then applied to the Nu river basin, a place with high altitude and complex topography and also with distinctive climate. From this perspective this study is meaningful. I would suggest that this paper be accepted with a few minor revisions.*

**Response**: We highly appreciate the reviewer's positive feedback. Our detailed responses are given after each comment (italics) below.

*Specific Comments:*

1) *Page-7, line 158, "59 gauges are available", which are not consistent with the number of stations in Figure 2.*

**Response**: We thank the reviewer for pointing out the inconsistency between the original Fig. 2 and text. We now have corrected Fig.2 with all the 59 gauges displayed, which is reproduced as follows:

[Figure]

**Figure 2** (a) Terrain map of the study area (the Nu-Salween basin and its adjacent areas), (b) The distribution of precipitation during the year across the Nu River.

2) *Page-7, line 164, it is suggested that a sentence be given to describe the temporal and spatial resolution of MOD13A3 and MYD13A3 vegetation products*

**Response**: The temporal and spatial resolutions of the MOD13A3 and MYD13A3 products are 1 month and 1 km, respectively. This description will be inserted into the revised manuscript in line 165, page 7.

3) *Page-8-9, line 208-209, "It is also noted that R2 values of RMIs for drier years are less than wetter years. …. ..". Time-lag effects of vegetation responses to precipitation can be considered to explain this phenomenon.*

**Response**: We thank the reviewer for the valuable suggestion. Although several studies indicate that vegetation responses to precipitation with a time lag varying from several days to 3 months according to different vegetation types, climates and latitudes (Wang et al., 2003; Bao et al., 2007; Long et al., 2010; Wu et al., 2015; Lin et al., 2015), such time-lag effects are not apparently observed in the monthly NDVI data we used in the Nu River application.

By comparing the annual NDVI with that of 1 month time lag (Fig. R1a), we see minimal difference between them with over 75% of the samples showing less than 1% difference (Fig. R1b). As such, we think that the time-lag effects of vegetation responses to precipitation are not capable of explaining smaller $R^2$ values of drier years than wetter years.

[Figure]

**Figure R1** (a) Comparison between yearly mean NDVI and yearly mean NDVI with 1 month lag, (b) frequency distribution of 1 month lag NDVI's relative change to NDVI without time lag

4) *Page-12, line 299, "the negative regression coefficient of temperature in RME+T indicates that precipitation decrease as the temperature increase". I don't agree that your explanation of negative*

*regression coefficient of temperature shows that that precipitation decreases as the temperature increase.*

**Response**: We thank the reviewer for pointing out the careless wording in this statement. Such statement was directly inferred from the negative regression coefficient of temperature to precipitation, with the aim to show the inconsistent trends of precipitation with temperature and elevation. We have reworded the statement in the revised manuscript as follows:

"The negative regression coefficient of temperature in RME+T indicates inconsistent trends between precipitation and temperature."

5) *Some detailed should be paid more attention. For example, line 191, correct website should be given; In Figure 2, legend of DEM should not be 0 in this region.*

**Response**: These details have been corrected/explained as follows:

1) The correct website address is added as:
   http://data.cma.cn/data/detail/dataCode/SURF_CLI_CHN_MUL_DAY_V3.0/keywords/v3.0.html.

2) In Figure 2, the bottom left corner is located in the Bay of Bengal where several few pixels have altitude value of zero.

**References:**

Bao, Y., Song, G., Li, Z., Gao, J., Lü, H., Wang, H., Cheng, Y. and Xu, T.: Study on the spatial differences and its time lag effect on climatic factors of the vegetation in the Longitudinal Range-Gorge Region, Chin. Sci. Bull., 52(2), 42–49, doi:10.1007/s11434-007-7005-5, 2007.

Brunsdon, C., McClatchey, J. and Unwin, D. J.: Spatial variations in the average rainfall-altitude relationship in Great Britain: an approach using geographically weighted regression, Int. J. Climatol., 21, 455–466, doi:10.1002/joc.614, 2001.

Lin, Y., Xin, X., Zhang, H. and Wang, X.: The implications of serial correlation and time-lag effects for the impact study of climate change on vegetation dynamics – a case study with Hulunber meadow steppe, Inner Mongolia, Int. J. Remote Sens., 36(19–20), 5031–5044, doi:10.1080/01431161.2015.1093196, 2015.

Long, H., Li, X., Bao, Y., Huang, L. and Li, Z.: Time lag analysis between vegetation and climate change in Inner Mongolia, in 2010 IEEE International Geoscience and Remote Sensing Symposium, pp. 1513–1516., 2010.

Wang, J., Rich, P. M. and Price, K. P.: Temporal responses of NDVI to precipitation and temperature in the central Great Plains, USA, Int. J. Remote Sens., 24(11), 2345–2364, doi:10.1080/01431160210154812, 2003.

Wu, D., Zhao, X., Liang, S., Zhou, T., Huang, K., Tang, B. and Zhao, W.: Time-lag effects of global vegetation responses to climate change, Glob. Change Biol., 21(9), 3520–3531, doi:10.1111/gcb.12945, 2015.

---

## Author Comment (AC2) · 23 Dec 2016

**Responses to Anonymous Referee #2:**

*General Comment:*

*This is a very interesting and clearly written paper on precipitation distribution in a mountainous river basin. Mountains, in particular high mountains in Asia, are the source of freshwater for many people, but the observation of precipitation is far from enough. For example, the first item of Scientific Questions of GEWEX under WCRP is the observation of precipitation. As such, the topic of this paper is within the scope of HESS, and is a hot topic in hydrology. As already described above, the paper is very clearly written, thus easy to follow. Therefore, I basically want to recommend this paper toward publication in HESS. Nevertheless, I would also mention that there are concerns, comments, or suggestions as written below.*

**Response**: We greatly appreciate the reviewer's positive feedback. Our detailed responses are given after each comment (*italics*) below.

*Specific Comments:*

1) *In the title, abstract and also in any other part of this paper, the authors should clearly mention that the target of this research is annual mean or climatological mean precipitation amount. The authors compared their product with IDW (of gauge-based precipitation) and a TRMM-based product. I can accept this product is better for annual mean and climatological mean. However, IDW and TRMM can provide us short timescale data, such as daily data. This is a major difference between these products against the product which the authors try to show in this paper.*

Response: We thank the reviewer for the valuable suggestion. We change the title as "Remapping annual precipitation in mountainous area based on vegetation pattern: a case study in Nu river" to emphasize the target of this research, and modify the corresponding parts in the manuscript.

2) *It is also recommended to clearly describe that this study is a case study for a specific area, in the title, abstract, and also in other places of this paper. Why I recommend so is described in the following. As shown in Figure 2 and Figure 6, in this target area, precipitation amount is larger for areas of lower elevation. In areas of lower elevation, it is usually expected that air temperature is warmer. It means, in the areas of lower elevation in this study, both precipitation and temperature are better, favorable, for NDVI. I suspect, this the reason why the authors can get a clear positive relationship between precipitation and NDVI. But, this is specific to this particular area. Of course, there could be other similar areas in the world. However, there must also be areas with different characteristics between NDVI and precipitation. Thus, the authors are requested to explicitly describe in the title and the abstract and also in relevant places of the main text that this study is a case study with such characteristics shown in Figures 2 and 6.*

**Response**: We thank the reviewer for the suggestion. We change the title as "Remapping annual precipitation in mountainous area based on vegetation pattern: a case study in Nu river" to emphasize the specific area of this research, and make it clear in the conclusion that comparison study is needed in other regions.

3) *This comment is a comment following the above comment. Very in general, precipitation is larger if elevation is higher. This is in contrast with what is seen in Figures 2 and 6. But, I need to add to "precipitation is larger if elevation is higher". Such a general tendency is probably true to 2000m or 3000m in elevation. In this regard, "we note that for simplicity, the extra determinants are assumed to have linear relationship with precipitation" is somewhat suspicious because this area has elevation up to 7000m. Is it possible to make a figure in which horizontal axis is elevation and vertical axis is precipitation (and NDVI) using observed precipitation data and NDVI data. There might be a positive relationship between elevation and precipitation up to 2000 or 3000m in elevation, and a negative or flat relationship between elevation and precipitation after it. Also, because NDVI favors large precipitation and warmer temperature (= lower elevation), the response of NDVI is different up to 2000 or 3000m and after it. NDVI-precipitation relation may depend on elevation bands such as lower than 3000m and higher than 3000m (of course, I do not know it would be 3000m or not which changes the relationship), but such an analysis was not done in this study as far as I can see. I think the authors can easily analyze.*

**Response**: The suggested analysis is conducted in two stages: 1) we examine the relationship between annual precipitation and elevation within different elevation ranges; and 2) we examine the relationship between annual precipitation and NDVI within corresponding elevation ranges as in stage 1, whose results are presented as follows:

    1)   the relationship between annual precipitation and elevation:

An overall negative relationship is found between precipitation and elevation for the whole elevation range 0–5000 m with a $R^2$ value of 0.62 (Fig. R2a), whereas there is only unapparent/weak relationship at different elevation bands (Fig. R2b-f). Given the spatial heterogeneity of orographic effects on precipitation (Brunsdon et al., 2001; Daly et al., 2008) and insufficient data of this study, a more thorough investigation of the relationship between precipitation and elevation needs to be conducted with more information that might be available in the future.

[Figure]

**Figure R1** The relationship between mean annual precipitation and elevation at different elevation bands, (a) whole elevation bands; (b) elevation band :<1000 m; (c) band:1000~2000 m; (d) band: 2000~3000 m; (e) band :3000~4000 m; (f) band: >4000 m.

2) the relationship between annual precipitation and NDVI:

Positive precipitation-NDVI relationships are found at different elevation bands (Fig. R3) with the best and worst fitness observed at elevation band 2000~3500 m with a $R^2$ value of 0.94 and at elevation band 0~2000 m with a $R^2$ value of 0.62, respectively. By comparing the three regressions at different bands with the global regression, we notice that more significant overestimates of precipitation are observed with the range of lower NDVI values (<0.4) at band 0–2000 m than other three regressions whereas regression at band >3500 m has an significant overestimation of precipitation than other three regressions for higher NDVI values(>0.5).

[Figure]

**Figure R2** The relationship between mean annual precipitation and NDVI at different elevation bands, (a) elevation band : <200m; (b) band: 2000~3500 m; (c) band: >3500 m; (d) whole bands; (e) comparison of precipitation-NDVI relationship for different bands .

To summarize, an overall negative relationship is found between precipitation and elevation across different elevations in the study region and the NDVI and precipitation demonstrates positive correlations at different elevation bands.

4) *Although target temporal and spatial scales are different, a recent study submitted to HESSD (Beck et al., 2016) provides a globally distributed precipitation data (called MSWEP) in which mountainous precipitation is corrected for gauge under-catch and also orographic effect was introduced by inferring catchment-average P from stream-flow (Q) observations at 13762 stations across the globe. I found mountainous precipitation is somehow well represented in the product by Beck after downloading the data from http://www.gloh2o.org and by making figures of the data by myself. Thus, it is recommended to have discussion in terms of Beck et al. (2016).*

**Response:** We thank the reviewer for the suggestion and present the comparison between MSWEP product and our product as follows:

Comparison in mean annual precipitation between the gauged measurements and predictions by the MSWEP (Multi-Source Weighted-Ensemble Precipitation, Beck et al. 2016) and TRMM product (Fig. R4) shows that TRMM and MSWEP predicted the precipitation well with an overall overestimation while RME product shows no obvious systematic deviation. The RMSE values for MSWEP, TRMM and RME are 241, 196 and 174 mm, respectively. The possible reason why MSWEP shows no superiority over TRMM in predicting annual precipitation is that few gauge data is available in this region which limited the efficiency of MSWEP method. However, the method in MSWEP does provide insights into the production of high temporal resolution (3-hourly) precipitation, which we believe will be helpful to our future work.

[Figure]

**Figure R4** Comparison in mean annual precipitation between the gauged measurements and predictions by the MSWEP, RMM and RME.

*5)   - A map of sub-basins is better to be provided (for Figure 10).*

**Response:** Added as suggested (Fig. R5).

[Figure]

**Figure R5** Sub-basins based on hydrologic stations

*6)   - I do not think Figure 7 is good to show. I can understand if the authors show the difference between Figure 8 and Figure 6. At least, I would say Figure 7 is awkward.*

**Response:** Removed as suggested.

*7)   - I also suspect that all the areas in Figures 6, 7, 8 are appropriate to show. I mean, there is almost no observation station in the left lower quarter of Figure 2. Then, I suspect whether the values of precipitation amount shown in Figure 6, 7, 8 for the left lower quarter of those figures are enough valid, particularly for the main product of this study and the IDW-based output.*

**Response:** We agree with that the values of precipitation shown in Figure 6, 7, 8 for the left lower quarter are doubtful because no observation station located in this region. Both RME and TRMM product show this region has large precipitation (>1800mm) and RME gives smaller values than TRMM. As discussed in our manuscript (Line 202-204), the regression model tends to underestimate precipitation as annual precipitation exceeds a certain threshold. We emphasized this conclusion in our manuscript and modified relevant figures.

*8)   Very finally, this is probably out of scope of this paper, but I am interested in whether major precipitation season is the same over this particular region. I mean, summer precipitation and winter precipitation (particularly solid precipitation like snow) may have different responses.*

**Response:** According to our previous study, the distributions of precipitation during the year varies significantly across this region. Fig. R6 shows the distributions of precipitation during the year for 7 stations located in the up, middle and downstream of Nu River. The upstream and downstream have similar distribution of precipitation with major precipitation occurs in summer and little occurs in winter

while the middle of Nu River has relatively large precipitation in winter and spring. The solid precipitation mainly occurs in upstream during winter and the amount is small.

[Figure]

**Figure R6** The distribution of precipitation during the year across the Nu River

References :

Brunsdon, C., McClatchey, J. and Unwin, D. J.: Spatial variations in the average rainfall-altitude relationship in Great Britain: an approach using geographically weighted regression, Int. J. Climatol., 21, 455–466, doi:10.1002/joc.614, 2001.

Daly, C., Halbleib, M., Smith, J. I., Gibson, W. P., Doggett, M. K., Taylor, G. H., Curtis, J. and Pasteris, P. P.: Physiographically sensitive mapping of climatological temperature and precipitation across the conterminous United States, Int. J. Climatol., 28, 2031–2064, doi:10.1002/joc.1688, 2008.

Beck, H.E., A.I.J.M. van Dijk, V. Levizzani, J. Schellekens, D.G. Miralles, B. Martens, A. de Roo (2016): MSWEP: 3-hourly 0.25 ° global gridded precipitation (1979–2015) by merging gauge, satellite, and reanalysis data, Hydrology and Earth System Sciences Discussions, doi:10.5194/hess-2016–236.

---

## Author Comment (AC3) · 23 Dec 2016

**Responses to Anonymous Referee #3:**

***General Comments:***

*Gridded precipitation data are very useful for hydrological application and others and ground-observation-based ones have been developed for many regions. This study investigated a methodology to develop gridded precipitation data for the Nu River basin based on the ground-observed precipitation combined with vegetation indices. As an interpolation method, the data fusion may be a strong tool especially for a sparse observation area. Its application to the Nu River with sparse observations may contribute to expanding a hydrological knowledge. This paper requires some more analyses to make readers more convinced of the effectiveness. Therefore, I recommend to revise this manuscript based on comments below before publication.*

**Response:** We thank the reviewer for the positive feedback. Our detailed responses are given after each comment (*italics*) below.

*Major comments:*

1) *The regression for RME uses all the data at the gauge stations and for the entire years. Such regression provide the climatological mean relationship between precipitation and NDVI. This regression cannot be applied for individual year as seen Table 1 where the coefficients distinct vary with year. Authors must mention the limitation of the proposed method in an appropriate paragraph.*

**Response:** We agree with the reviewer that the limitation in applying the RME regression to each year's data needs to be appreciated, which has been added in Section 5 of the revised manuscript as follows:

> In addition, although the RME model can utilize the full knowledge of precipitation in the entire study period compared with RMI models, the difference in the coefficients suggests apparent inter-annual variability of precipitation that should be considered when applying these models. Given the duration of study period and purpose, we suggest the RME model be used for long-term climatology identification while RMI models for inter-annual variability examination.

2) *Judging from Figure 6, climatological mean annual precipitation seems to depend on elevation. The dependence of precipitation on elevation is a well-known fact. In order to clarify the effectiveness of your method, it is better to compare the geographical distributions of climatological mean precipitation between your methods and a method by a regression between precipitation and elevation. A figure of the difference between the two may provide an important suggestion about strong points of your method.*

**Response:** We thank the reviewer for the insightful suggestion. In the revised manuscript, we compare our method (i.e., RME) with the precipitation-elevation regression (DEMP) model and present the difference of their precipitation estimates as the new Figure 9, which is reproduced as follows:

[Figure]

**Figure 9** (a) The map of precipitation estimates of DEMP; (b) difference in precipitation estimates between RME and DEMP.

***Specific Comments:***

3)  *L66: The objective of this study should be more specified in terms of spatio-temporal scale: climatological annual mean and 1 km.*

**Response:** Added in the abstract and introduction as suggested.

4)  *L103: The center of a certain box is not used in computing the averaged value of the grid box? If so, please provide a reason.*

**Response:** We thank the reviewer for pointing out incorrect statement. The center of a certain box is used in calculating the averaged value of the grid box. Related statement is corrected in the revised manuscript.

5)  *L157: The number of rain gauge stations in the Nu River basin seems smaller than 13 as long as it is judged from Figure 2.*

6)  *L158: Moreover, the 59 stations are not plotted on Figure 2.*

**Response:** We thank the reviewer for pointing out the inconsistency between the original Fig. 2 and text. We now have corrected Fig.2 with all the 59 gauges displayed, which is reproduced as follows:

[Figure]

**Figure 2** (a) Terrain map of the study area (the Nu-Salween basin and its adjacent areas). (b) The distribution of precipitation during the year across the Nu River.

7) *L159: Please explain what you mean by "climatic and topographic conditions are consistent with the Nu River basin."*

**Response:** We mean the enlarged area has similar climatic and topographic conditions as the Nu River basin: both regions are characterized as mountainous areas under the subtropical climate influenced by southeast and southwest monsoons. Such explanation has been added in the revised manuscript.

8) *L162: You use three terminologies: uncertainty, reliability, and accuracy in the 3.2.1 subsubsection. Please provide short definitions if you distinguish them in this manuscript.*

**Response:** We are sorry about bringing up the confusion due to the inconsistency in our terminology. We now only use "reliability" across Section 3.2.1 in the revised manuscript.

9) *L163: In this paragraph, please provide the spatio-temporal resolution of the MODIS dataset.*

**Response:** The temporal and spatial resolutions of the MOD13A3 and MYD13A3 data we used are 1 month and 1 km, respectively. This information has been added in the revised manuscript.

10) *L176: "m" in "merged" should be in standard font.*

**Response:** Corrected as suggested.

11) *L185: How do you classify an upscaled 1 km pixel when it is composed of two forest and two cropland pixels? You do not consider the mixed pixel? If so, please provide this information.*

**Response:** If any of the four 500 m pixels in MCD12Q1 classified as water, urban, snow or ice and cropland, the upscaled 1 km pixel will be classified as abnormal pixel (or non-natural vegetation) and assigned with a missing value (i.e. -9999), otherwise will be classified as normal pixel (or natural vegetation) and assigned with a 1 value.

12) *L196: Readers may know the rationale about the selection of the regression form. Not by "not shown here" but "judging from Figure 3" is better.*

**Response:** We thank the reviewer for the suggestion. Figure 3 is modified to include the comparison of the four regression forms and updated in the revised manuscript.

[Figure]

**Figure 3** (a) Different regression forms for precipitation –NDVI relationship; (b) The precipitation-NDVI relationships for RME and RMI

13) *L202: Did you use these results when you draw Figure 6? If not, why don't you use these important information?*

**Response:** We did use these results in making the original Figure 6 (now Figure 8 in the revised manuscript). However, we didn't make any correction to the pixels out of the range from 400 mm to 1500 mm because there is no justifiable methods for such correction. Given the limited fraction of invalid pixels (10% in the whole study area and 7% in the Nu River basin), we still have them plotted in the Figure 8 to demonstrate a full picture of the spatial precipitation pattern in the study area, but we note those pixels are of large uncertainties and should be interpreted with caution.

*14) L220: Two spaces exist between of and regression.*
**Response:** The redundant space is removed.

*15) L222: Scenario should be Scenarios.*
**Response:** Corrected as suggested.

*16) L242: Please provide rationale about the use of the IDW method here.*
**Response:** IDW is one of the most popular methods for spatial interpolation of precipitation due to its easy implementation and flexibility in incorporating other auxiliary information (e.g., elevation).Such statement has been added in the revised manuscript.

*17) L264: Precipitation by the RME method often the largest among the three in Figure 10 is reasonable? Can you validate this precipitation estimates larger than the other two by sub-basin water balance such as observed river discharge = P – E. In other words, large amount of precipitation is better than small one in order to explain the observed river discharges.*
**Response:** We thank the reviewer for this suggestion. However, we deem that it would still be difficult to justify the magnitude order of estimates by the three methods even if the observed river discharge of a certain sub-basin is provided: the ***observed*** river discharge implies the response of a basin to the only ***realistic*** precipitation rather than different ***estimates***. In other words, it is difficult to infer the impacts of different inputs (i.e., precipitation estimates by different methods) based on a single output (i.e., river discharge observation).

To evaluate the accuracy of different precipitation estimates, we utilize MODIS evapotranspiration products MOD16 to calculate the water balance based precipitation (i.e. ET+R). Then we compare it with 5 precipitation products and the results are presented in Fig. R2. DEMP represents precipitation based on precipitation-elevation relationship, BandP represents precipitation based on precipitation-NDVI relationship with consideration of elevation band. It can be found that RME and BandP produce closer estimation to water balance based precipitation, implying that the precipitation mapping result based on precipitation-NDVI relationship is reasonable.

[Figure]

**Figure R2** Comparison between water balance based precipitations (R+ET) and 5 precipitation products: DEMP (P-elevation relationship), BandP (P-NDVI relationship with consideration of elevation band), RME, TRMM and IDW. Here GS, JC, GLH, DWJ and LK-GS stand for Gongshan, Jiuchen, Gulaohe, Dawanjing and Liuku-Gongshan, respectively.

18) *L295: As mentioned above, in addition to RME+T and RME+H, how the regression of T or H onto precipitation works for producing climatological mean annual precipitation like Figure 6.*

**Response:** Our intention of using RME+T and RME+H was to demonstrate the inconsistent trends of precipitation with temperature and elevation. According to Table 5 and Fig. 15, the differences in performance metrics and the regression coefficients between RME+T, RME+H and RME are minimal. Therefore, we think that the influence of including H and T on the regression results is limited.

[Figure]

**Figure 15** Spatial precipitation difference between RME and (a) RME+H; (b)RME+T

*19) Figures 1: Font color should be the same as in outline color of boxes.*

*20) Figures 2, 6, 7, 8, and A2: Minutes and seconds should be removed from the annotations of the coordinates. Font size should be enlarged so as to see them easily.*

**Response:** Modified as suggested.

*21) Figure 4: Please provide explanations about two symbols in the figure caption.*

**Response:** We thank the reviewer for this suggestion .The triangle markers denote the values (R, $R^2$ and RMSE) of RME model. The plus markers represent the outliers that are out of the range from (Q1-1.5IQR) to (Q3+1.5IQR). Q1 and Q3 are the 25[th] and 75[th] percentiles, and IQR (=Q3-Q1) is the interquartile range. Such explanation has been added in the revised manuscript.

---

## Author Comment (AC4) · 23 Dec 2016

[revised manuscript text omitted]
 ($E_{RMS}$), mean relative error ($E_{MR}$) and mean absolute relative error ($E_{MAR}$), which are given as follows:

[revised manuscript text omitted]

and 91 °E–101 °E is chosen for the application of the fusion framework, where 59 gauges are available and the climatic and topographic conditions are similar: both regions are characterized as mountainous areas under the subtropical climate influenced by southeast and southwest monsoons. Besides, given no rain gauges are available outside of China in this study region, the non-Chinese region is excluded from the study area.

**3.2 Datasets**

**3.2.1 Vegetation data**

In this study, we use two MODIS (moderate resolution imaging spectoradiometer) vegetation products,

MOD13A3 (termed MOD hereafter) and MYD13A3 (termed MYD hereafter), in the application of the fusion framework. Both the MOD and MYD datasets contain 10 sub-datasets consisting of NDVI, EVI and pixel reliability. The temporal and spatial resolutions of the MOD13A3 and MYD13A3 products are 1 month and 1 km, respectively. The pixel reliability is an accuracy metric of the data quality pixel and has four valid values: 0 for good reliability, 1 for marginal reliability, 2 for snow/ice, and 3 for cloud. Based on the pixel reliability information, the NDVI values are either selected for corresponding pixel reliability levels being 0 and 1 or discarded as anomalies otherwise.

The MOD dataset is used as benchmark while MYD is taken as the alternative for occasions when MOD data are missing or have large uncertainties. Since both the MOD and MYD datasets are extracted from different satellites at different transit times, systematic errors may exist in the difference between the two datasets. As such, we construct two regressions to remove their systematic errors: one is based on a subset with both MOD and MYD

of good reliability (= 0), and the other on a subset with MOD of marginal reliability (= 1) and MOD of good reliability (= 0). After the removal of systematic errors, a merged dataset of MOD and MYD (termed MMD

hereafter) is generated under the criteria given as follows:

$$MMD = \begin{cases} MOD & (MOD == 0) \\ MYD & (MOD > 1 \ \& \ MYD == 0) \\ MOD & (MOD == 1 \ \& \ MYD == 1) \\ NULL & (MOD > 1 \ \& MYD > 0) \end{cases} \qquad (6)$$

The annual MMD dataset is then calculated by averaging the 12 monthly images.

**3.2.2 Landuse data**

The landuse dataset MCD12Q1 Version 51 (MODIS/Terra+Aqua Land Cover Type Yearly L3 Global 500m SIN Grid V051) in period of 2001-2013 is used to identify the anomalies of MMD, while the IGBP (International Geosphere Biosphere Programme) classification is adopted for its wide applications. Due to mismatch in spatial resolutions between MMD and MCD12Q1 datasets, the MCD12Q1 dataset is upscaled to 1km as MMD for anomaly identification. It should be noted that if any of the four 500 m pixels in MCD12Q1 classified as water, urban, snow/ice and cropland, the upscaled 1 km pixel will be classified as anomalous pixel (or non-natural vegetation) and assigned with a missing value (i.e. -9999), otherwise will be classified as normal pixel (or natural vegetation) and assigned with a 1 value.

**3.2.3 Weather data**

Datasets consisting of daily precipitation and air temperature collected at the 59 gauges in the study area are obtained via the China Meteorological Data Sharing Service system (http://data.cma.cn/data/detail/dataCode/SURF_CLI_CHN_MUL_DAY_V3.0/keywords/v3.0.html).
The air temperature measurements will be used for dependence analysis later in Section 4.5. The streamflow data provided by Yunnan University will be used for calculating sub-basin scaled precipitation based on water balance. The 5 hydrological stations are Gongshan, Liuku, Jiucheng, Gulaohe and Dawanjiang with the drainage area of 101146, 106681, 6308, 4185 and 7986 km$^2$, respectively. MODIS evapotranspiration (ET) product MOD16 (http://www.ntsg.umt.edu/project/mod16) with the spatiotemporal resolution of 1 km/1 weekly will also be used in calculating precipitation based on water balance.

**4 Results and discussion**

**4.1 Model calibration and validation**

Based on the results of six evaluation metrics for different regression form candidates (Fig. 3a), the 2$^{nd}$-order polynomial is chosen as the regression model form in this study:

$$p = a \, NDVI^2 + b \, NDVI + c \tag{7}$$

where $p$ denotes precipitation amount in mm, and $a$, $b$ and $c$ are regression coefficients. The results of regression coefficients and evaluation metrics are given in Table 1, and the precipitation-NDVI relationships for the study period are demonstrated in Fig. 3b.

The best performance of the regression model is found within $0.2 < NDVI < 0.7$ and 400 mm year$^{-1}$ < p < 1500

mm year$^{-1}$. Larger errors are found at pixels with NDVI larger than 0.7 or annual precipitation larger than 1500

mm, implying the water supply is no longer a determinant of vegetation growth as annual precipitation exceeds a certain threshold.

In general, the RMIs demonstrate better performance than RME, which can be attributable to the less variability of precipitation in a single year than the whole study period. It is also noted that the $R^2$ values of RMIs for drier years (2003, 2009 and 2011) are less than wetter years, indicating the weaker coupling effect between vegetation growth and precipitation.

The performance of regression models is assessed under three scenarios as described in Section 2.2. A total of 300

tests are conducted and performance metrics (i.e., R, $R^2$, $E_{RMS}$, and $E_{MAR}$) are calculated accordingly (Fig. 4 and

Table 2). The high R values (> 0.85) indicate a strong correlation between NDVI and precipitation independent of sampling method. Also, the regression models demonstrate good performance with $R^2$ larger than 0.75 and

$E_{MAR}$ less than 20%. In addition, the metrics of regression models fluctuate around that of the RME with narrow inter-quartile ranges, indicating the regression models have remarkable consistency with the RME model.

Scenario a is designed to examine inter-annual stability in the performance of regression models, where the good performance indicates the acceptable ability of the RME model in estimating precipitation during periods when precipitation measurements are not available. Scenarios b and c investigate the impacts of spatial and temporal coverages of measurements, respectively. It is noteworthy that under scenario b better performance in regression models is observed as compared with scenario c, implying greater importance of spatial coverage of measurements in conducting the regressions. In addition, the results of calibration is better than validation as revealed by all metrics criterions as expected. However, the differences between calibration and validation are not significant, implying the consistent performance of regression models under various scenarios.

The performance of RME is further assessed by comparing the estimates against observations (Fig. 5), and good agreement between estimates and observations is observed. It should be noted the RME shows difficulty in estimating precipitation larger 2000 mm ( cf. the dashed line in Fig. 5), implying the limitation of the fusion framework inherited from the oversaturation effect of vegetation index.

Elevation effect on the relationship between precipitation and NDVI is a concern to appreciate. An overall negative relationship is found between precipitation and elevation for the whole elevation range (i.e., 0–5000 m)

with a $R^2$ value of 0.62 (Fig. 6a), whereas there is only unapparent/weak relationship at different elevation bands (Fig. 6b-f). Given the spatial heterogeneity of orographic effects on precipitation (Brunsdon et al., 2001; Daly et al., 2008) and insufficient data of this study, a more thorough investigation of the relationship between precipitation and elevation needs to be conducted with more information that might be available in the future.

Positive precipitation-NDVI relationships are found at different elevation bands (Fig. 7) with the best and worst fitness observed at elevation band 2000–3500 m with a $R^2$ value of 0.94 and at elevation band 0–2000 m with a

$R^2$ value of 0.62, respectively. By comparing the three regressions at different bands with the global regression, we notice that more significant overestimates of precipitation are observed with the range of lower NDVI values (<0.4) at band 0–2000 m than other three regressions whereas regression at band >3500 m has a significant overestimation of precipitation than other three regressions for higher NDVI values (>0.5).

**4.2 Spatial characteristics of precipitation**

The spatial characteristics of precipitation of the study area are investigated with RME for the whole study period (Fig. 8). Annual precipitation in Nu River is observed to decrease from south to north and from west to east with prominent spatial variability. Two "hot-spot" regions, whose annual precipitation exceeds 1500 mm, can be identified in the study areas: one near south border and the other close to southwestern mountain border. The east part of the Nu river basin featuring a dry and warm climate receives an average annual precipitation of 800 mm with large inter-annual variability. A precipitation product based on precipitation-elevation relationship (DEMP)

is used to compare with RME. There is no obvious distribution pattern of precipitation (Fig.9a) and a smaller spatial variability compared to RME in the DEMP product, indicating the advantage of RME in representing the spatial variability of annual precipitation. And the overall underestimation of precipitation is observed in the

DEMP product across the whole study area (Fig.9b). In addition, the pixels in Fig.8 with a value out of the valid range (i.e., 400 mm/yr < P< 1500 mm/yr) may have relatively large error as discussed in section 4.1. As there is no justifiable methods for such correction and given the limited fraction of invalid pixels (10% in the whole study area and 7% in the Nu River basin), the figure can be used to demonstrate a full picture of the spatial precipitation pattern in the study area, but we note those pixels are of large uncertainties and should be interpreted with caution.

**4.3 Model performance comparison**

The performance between IDW approach, TRMM product and the fusion framework is compared in this section.

IDW is one of the most popular methods for spatial interpolation of precipitation due to its easy implementation and flexibility in incorporating other auxiliary information (e.g., elevation). In general, the IDW approach is unable to demonstrate the high spatial variability though it can capture the general spatial distribution of whole basin (Fig. 10a) as TRMM (Fig. 10b). Due to the coarse spatial resolution, TRMM cannot capture the high variability in the river valley where the elevation varies significantly. Although large precipitation (>1800mm) is observed in both our and TRMM products in the southwest of the study area region, our product gives lower precipitation compared to TRMM. As discussed above, the regression model tends to underestimate precipitation as the annual precipitation exceeds a certain threshold because the water supply is no longer a determinant of vegetation growth.

To demonstrate the advantage of the fusion framework, a cross-validation is conducted against the randomly sampled gauge observations by varying the number of samples (1 - 40). The cross-validation shows higher $E_{RMS}$

for the IDW approach, followed by TMMM and RME (Fig. 11a). A higher mean $E_{MR}$ of 15% is observed for

TRMM than IDW (8%) and RME (5%) while the difference in $E_{MAR}$ are minimal between TRMM and IDW. The results indicate an overestimated precipitation by TRMM as compared to gauge observations. Table 3 summarizes the maximum, minimum and mean values of each method and shows the relative difference between RME and other two methods. On average, $E_{RMS}$ of RME is smaller than that of IDW and TRMM by 20.4% and 17.4%, respectively. In general, the fusion framework demonstrates better performance than the other approaches.

To further evaluate the performance of RME, the annual averages of precipitation of five hydrological stations (Fig. 12a) and whole basin estimated by the three approaches (IDW, RME and TRMM) are compared. At the whole basin scale, the estimate by RME is 5.2% higher than that of IDW while 7.9% lower than TRMM. Although the difference between the three approaches is minimal at the basin scale, the difference at the sub-basin scale is remarkable. In the upstream region (i.e., Gongshan sub-basin) located in Tibet Plateau, TRMM overestimates precipitation by 13.2% while IDW underestimates by 7.6% as compared with RME. In the other four downstream sub-basins, estimates by RME are larger than those by IDW and TRMM. In general, in the midstream and downstream regions with large variability in terrain height, RME gives larger estimates of precipitation than IDW

and TRMM.

To evaluate the accuracy of different precipitation estimates, we utilize MODIS evapotranspiration product

MOD16 to calculate water balance based precipitation (i.e. ET+R). Then we compare it with 5 precipitation products including RME, BandP (precipitation based on precipitation-NDVI relationship with consideration of elevation band), DEMP, TRMM, IDW (Fig.12b). It can be found that RME and BandP produce closer estimation to water balance based precipitation, implying that the precipitation mapping result based on precipitation-NDVI

relationship is reasonable.

We also compared our products with the Multi-Source Weighted-Ensemble Precipitation (MSWEP) product. The dataset takes advantage of a wide range of data sources, including gauges, satellites, and atmospheric reanalysis models, to obtain the best possible precipitation estimates at global scale with a high 3-hourly temporal and 0.25 °

spatial resolution (Beck et al., 2016). Comparison in the annual mean precipitation between the gauge measurements and predictions by the MSWEP and TRMM product (Fig. 13) shows acceptable performance of both MSWEP and TRMM in predicting the precipitation with an overall overestimation. The RMSE values for

MSWEP, TRMM and RME are 241 mm, 196 mm, and 174 mm, respectively, indicating that RME gives the best prediction among the three products. The possible reason why MSWEP shows no superiority over TRMM in predicting annual precipitation is that very few gauges are available in this region that might limit the applicability of MSWEP method. However, the MSWEP method does provide insights into the production of high temporal resolution (3-hourly) precipitation, which we believe will be helpful to our future work.

**4.4 influence of different vegetation index**

Considering the possible degradation in model performance caused by oversaturation of NDVI in high biomass areas, another vegetation indicator, Enhanced Vegetation Index (EVI), is suggested as an alternative for estimating vegetation growth (Matsushita et al., 2007; Liao et al., 2015). As such, we also test the fusion framework with

EVI in addition to NDVI and the results are assessed against the gauge observations.

Based on the chosen metrics, EVI is found to outperform NDVI with better regression quality (Table 4): EVI- based regression model gives higher $R^2$, smaller $E_{RMS}$ and $E_{MAR}$ compared to the NDVI-based model. Also, remarkable difference is observed in the precipitation estimates based on the two vegetation indices (Fig. 14). It is noted that the curvature of EVI-based model is larger than NDVI-based model, suggesting higher sensitivity of

EVI-based model in humid environment. Although the EVI-based model demonstrates better performance than the NDVI-based one, it should be noted that NDVI is the most popular vegetation index used in operational applications among the available vegetation index products. Besides, NDVI has a relative longer temporal coverage compared to other vegetation index products. For instance, the AVHRR (Advanced Very High

Resolution Radiometer) NDVI data are available since 1982 with a global coverage. As such, under scenarios when EVI is unavailable, NDVI is a satisfactory index that can be used in the fusion framework.

**4.5 Influence of other ambient determinants**

One major assumption of the proposed framework is that precipitation is the only determinant of vegetation growth and thus NDVI is regarded as a proxy for precipitation. However, other ambient factors, such as soil properties, solar radiation, air temperature, elevation, etc., may significantly influence the vegetation growth as well as NDVI values. Considering the data availability of various ambient factors, air temperature and elevation, in addition to NDVI, are adopted as extra determinants to establish the regression models, which are thus termed as RME+T and RME+H for air temperature and elevation, respectively. We note that for simplicity, the extra determinants are assumed to have linear relationship with precipitation.

The difference in $R^2$, $E_{RMS}$, and $E_{MAR}$ between the three models are minimal and the regression coefficients of the three models are very close to each other (Table 5). The negative regression coefficient of temperature in RME+T

indicates inconsistent trends between precipitation and temperature. Since the temperature decreases with the increase in elevation, RME+T and RME+H essentially provides consistent estimates of precipitation which is also clearly shown in Fig. 15. It is also noted the added information by extra determinants (i.e., air temperature and elevation) is in fact minimal. Overall there is little difference between RME and other two products. Therefore, we consider the RME-only based vegetation index as a simple and efficient model for precipitation estimation.

**5 Conclusion**

In this study, a satellite-gauge-vegetation fusion framework has been developed for estimating the precipitation in mountainous areas by establishing regression relationship between gauge-based precipitation observations and satellite-based vegetation dataset. The fusion framework was then applied in the Nu River basin of Southwest

China for estimating precipitation between 2001 and 2012.

The fusion framework for the Nu River basin adopted a second order polynomial form and demonstrated promising ability in capturing the high spatial variability of precipitation in the river valley. Six evaluation metrics, including R, $R^2$, $E_{RMS}$, $E_{MR}$ and $E_{MAR}$, indicated good performance of the fusion framework in precipitation estimation. The performance of the fusion framework was also compared with the IDW approach and TRMM

product and the comparison results indicated that the fusion framework generally outperformed other approaches in estimating precipitation in mountainous areas. On average, the $E_{RMS}$ of the fusion framework is 20.4%, 17.4%

smaller than that of IDW and TRMM, respectively. $E_{MR}$ of the fusion framework is 1.2%, 71.5% smaller than that of IDW and TRMM. $E_{MAR}$ the fusion framework is 18.9%, 28.3% smaller than that of IDW and TRMM.

The success of application of the fusion framework in the Nu River sheds light on the precipitation estimation in mountainous areas by using multi-source datasets. However, this framework does have certain limitations that are important to appreciate. First, the framework is applied only in the Nu River basin. More mountainous areas under different climates need to be examined to further test the robustness of this framework. In addition, although the

RME model can utilize the full knowledge of precipitation in the entire study period compared with RMI models, the difference in the coefficients suggests apparent inter-annual variability of precipitation that should be considered when applying these models. Given the duration of study period and purpose, we suggest the RME

model be used for long-term climatology identification while RMI models for inter-annual variability examination.

Also, to fully verify the theoretical basis of this framework that vegetation actively interacts with precipitation in mountainous areas, future work is required to refine the spatiotemporal resolution of this study to enable better scrutiny into vegetation-precipitation interactions at sub-monthly scales across more detailed vegetation species.

*Acknowledgments*

The study is supported by NSFC under grant U1202231, 51679119 and 91647107, National Key Technology

Support Program under grant 2011BAC09B07-3 and by China Postdoctoral Science Foundation under grant

2015T80093. The authors thank China Meteorological Administration, Yunnan University, MODIS NDVI,

Tropical Rainfall Measuring Mission (TRMM) and the Shuttle Radar Topography Mission (SRTM) for providing the data used in this study.

**Appendix: Merging of NDVI datasets**

The merging of NDVI datasets improves the accuracy as expected (Fig. A1), the monthly error rates (i.e., the ratio of the pixel which quality value is over 1) of MOD and MMD are generally reduced with an average of 5% and over 20% in several months. Fig.A2 shows that the accuracy of MMD is significantly improved in a ridge area covering 23 °10′ N–23 °40′ N and 98 °30′ E–99 ° E. Fig. A2b shows NDVI value near right and left boundary is underestimated by MOD. Fig.A2c shows NDVI value in the middle boundary is underestimated by MYD. The underestimates in both products near the boundary of MOD and MYD are amended (Fig. A2a). Fig.A3 shows the three NDVI series for one rain gauge. Comparing with MOD series, the improved accuracy in MMD is mainly observed in the wet season (from May to October), when the NDVI values could be often underestimated due to the overcasts.

[revised manuscript text omitted]

relationships for RME and RMI.

[Figure]

**Figure 4** Box plots of R, $R^2$, $E_{RMS}$ of RME model under three scenarios: a) fully random; b) all gauges, partial period; and c) partial gauges, entire period. Details of the three scenarios refer to Section 2.2. The triangle markers denote the value (R, R2 and RMSE) of RME model. The plus markers represent the outliers that are out of the range from (Q1-1.5IQR) to (Q3+1.5IQR). Q1 and Q3 are the 25th and 75th percentiles, and IQR (=Q3-Q1) is the interquartile range.

[Figure]

**Figure 5** Comparison in annual precipitation between the gauged measurements and predictions by the regression model for scenario a) fully random; b) all gauges, partial period; and c) partial gauges, entire period.

Details of the three scenarios refer to Section 2.2.

[Figure]

**Figure 6** The mean annual precipitation-elevation relationships at different elevation bands, (a) whole elevation band; (b) elevation band :<1000 m; (c) band:1000~2000 m; (d) band: 2000~3000 m; (e) band :3000~4000 m; (f)

band: >4000 m.

[Figure]

**Figure 7** The mean annual precipitation-NDVI relationships at different elevation bands, (a) elevation band : <200

m; (b) band: 2000~3500 m; (c) band: >3500 m; (d) whole elevation band; (e) comparison of precipitation-NDVI

relationship at different bands.

[Figure]

**Figure 8** Average annual precipitation distribution of 2003-2012 from RME.

[Figure]

**Figure 9** (a) The map of precipitation estimates of DEMP; (b) difference in precipitation estimates between
RME and DEMP.

[Figure]

**Figure 10** Spatial distribution of mean annual precipitation of 2003-2012 estimated by (a) IDW and (b) TRMM.

[Figure]

**Figure 11** Performance of $E_{RMS}$, $E_{MR}$ and $E_{MAR}$ for three methods in different remove numbers.

[Figure]

**Figure 12** (a) Sub-basins based on hydrological stations (b) Comparison between water balance based precipitations (R+ET) and 5 precipitation products: DEMP (P-elevation relationship), BandP (P-NDVI relationship with consideration of elevation band), RME, TRMM and IDW. Here GS, JC, GLH, DWJ and LK-GS stand for Gongshan, Jiuchen, Gulaohe, Dawanjing and Liuku-Gongshan, respectively.

[Figure]

**Figure 13** Comparison in mean annual precipitation between the gauged measurements and predictions by the MSWEP, RMM and RME.

[Figure]

**Figure 14** Regression relationship between annual precipitation and normalized NDVI/EVI

[Figure]

**Figure 15** Spatial precipitation difference between RME and (a) RME+H; (b) RME+T.

[Figure]

**Figure A1** Monthly Error rate of MOD, MYD and MMD

[Figure]

**Figure A2** Comparison of three NDVI products over a ridge area on June 2006, (*a*) for MMD, (*b*) for MOD, (*c*)

for MYD

[Figure]

**Figure A3** Comparison of three NDVI monthly time series over one gauge